# Non-monotonic Resource Utilization in the Bandits with Knapsacks Problem

**Raunak Kumar**
Department of Computer Science
Cornell University
Ithaca, NY 14853
raunak@cs.cornell.edu

**Robert D. Kleinberg**
Department of Computer Science
Cornell University
Ithaca, NY 14853
rdk@cs.cornell.edu

## Abstract

Bandits with knapsacks (BwK) [6] is an influential model of sequential decision-making under uncertainty that incorporates resource consumption constraints. In each round, the decision-maker observes an outcome consisting of a reward and a vector of nonnegative resource consumptions, and the budget of each resource is decremented by its consumption. In this paper we introduce a natural generalization of the stochastic BwK problem that allows non-monotonic resource utilization. In each round, the decision-maker observes an outcome consisting of a reward and a vector of resource *drifts* that can be positive, negative or zero, and the budget of each resource is incremented by its drift. Our main result is a Markov decision process (MDP) policy that has *constant* regret against a linear programming (LP) relaxation when the decision-maker *knows* the true outcome distributions. We build upon this to develop a learning algorithm that has *logarithmic* regret against the same LP relaxation when the decision-maker *does not know* the true outcome distributions. We also present a reduction from BwK to our model that shows our regret bound matches existing results [14].

## 1 Introduction

Multi-armed bandits are the quintessential model of sequential decision-making under uncertainty in which the decision-maker must trade-off between exploration and exploitation. They have been studied extensively and have numerous applications, such as clinical trials, ad placements, and dynamic pricing to name a few. We refer the reader to Bubeck and Cesa-Bianchi [7], Slivkins [17], Lattimore and Szepesvári [13] for an introduction to bandits. An important shortcoming of the basic stochastic bandits model is that it does not take into account resource consumption constraints that are present in many of the motivating applications. For example, in a dynamic pricing application the seller may be constrained by a limited inventory of items that can run out well before the end of the time horizon. The bandits with knapsacks (BwK) model [18, 19, 5, 6] remedies this by endowing the decision-maker with some initial budget for each of $m$ resources. In each round, the outcome is a reward and a vector of nonnegative resource consumptions, and the budget of each resource is decremented by its consumption. The process ends when the budget of any resource becomes nonpositive. However, even this formulation fails to model that in many applications resources can get replenished or renewed over time. For example, in a dynamic pricing application a seller may receive shipments that increase their inventory level.

**Contributions** In this paper we introduce a natural generalization of BwK by allowing non-monotonic resource utilization. The decision-maker starts with some initial budget for each of $m$ resources. In each round, the outcome is a reward and a vector of resource *drifts* that can be positive, negative or zero, and the budget of each resource is incremented by its drift. A negative drift has

36th Conference on Neural Information Processing Systems (NeurIPS 2022).

the effect of decreasing the budget akin to consumption in BwK and a positive drift has the effect of increasing the budget. We consider two settings: (i) when the decision-maker *knows* the true outcome distributions and must design a Markov decision process (MDP) policy; and (ii) when the decision-maker *does not know* the true outcome distributions and must design a learning algorithm.

Our main contribution is an MDP policy, `ControlBudget(CB)`, that has *constant* regret against a linear programming (LP) relaxation. Such a result was not known even for BwK. We build upon this to develop a learning algorithm, `ExploreThenControlBudget(ETCB)`, that has *logarithmic* regret against the same LP relaxation. We also present a reduction from BwK to our model and show that our regret bound matches existing results.

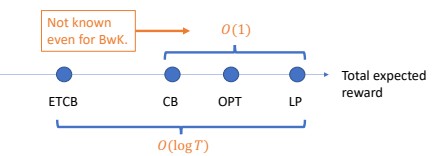

Instead of merely sampling from the optimal probability distribution over arms, our policy samples from a perturbed distribution to ensure that the budget of each resource stays close to a decreasing sequence of thresholds. The sequence is chosen such that the expected leftover budget is a constant and proving this is a key step in the regret analysis. Our work combines aspects of related work on logarithmic regret for BwK [9, 14].

**Related Work**  Multi-armed bandits have a rich history and logarithmic instance-dependent regret bounds have been known for a long time [12, 3]. Since then, there have been numerous papers extending the stochastic bandits model in a variety of ways [4, 16, 11, 6, 10, 2, 1].

To the best of our knowledge, there are three papers on logarithmic regret bounds for BwK. Flajolet and Jaillet [9] showed the first logarithmic regret bound for BwK. In each round, their algorithm finds the optimal basis for an optimistic version of the LP relaxation, and chooses arms from the resulting basis to ensure that the average resource consumption stays close to a pre-specified level. Even though their regret bound is logarithmic in $T$ and inverse linear in the suboptimality gap, it is exponential in the number of resources. Li et al. [14] showed an improved logarithmic regret bound that is polynomial in the number of resources, but it scales inverse quadratically with the suboptimality gap and their definition of the gap is different from the one in Flajolet and Jaillet [9]. The main idea behind improving the dependence on the number of resources is to proceed in two phases: (i) identify the set of arms and binding resources in the optimal solution; (ii) in each round, solve an adaptive, optimistic version of the LP relaxation and sample an arm from the resulting probability distribution. Finally, Sankararaman and Slivkins [15] show a logarithmic regret bound for BwK against a *fixed-distribution* benchmark. However, the regret of this benchmark itself with the optimal MDP policy can be as large as $O(\sqrt{T})$ [9, 14].

## 2 Preliminaries

### 2.1 Model

Let $T$ denote a finite time horizon, $\mathcal{X} = \{1, \ldots, k\}$ a set of $k$ arms, $\mathcal{J} = \{1, \ldots, m\}$ denote a set of $m$ resources, and $B_{0,j} = B$ denote the initial budget of resource $j$. In each round $t \in [T]$, if the budget of any resource is less than 1, then $\mathcal{X}_t = \{1\}$. Otherwise, $\mathcal{X}_t = \mathcal{X}$. The algorithm chooses an arm $x_t \in \mathcal{X}_t$ and observes an outcome $o_t = (r_t, d_{t,1}, \ldots, d_{t,m}) \in [0, 1] \times [-1, 1]^m$. The algorithm earns reward $r_t$ and the budget of resource $j \in \mathcal{J}$ is incremented by drift $d_{t,j}$ as $B_{t,j} = B_{t-1,j} + d_{t,j}$.

Each arm $x \in \mathcal{X}$ has an outcome distribution over $[0, 1] \times [-1, 1]^m$ and $o_t$ is drawn from the outcome distribution of the arm $x_t$. We use $\mu_x^o = (\mu_x^r, \mu_x^{d,1}, \ldots, \mu_x^{d,m})$ to denote the expected outcome vector of arm $x$ consisting of the expected reward and the expected drifts for each of the $m$ resources.[1] We also use $\mu^{d,j} = (\mu_x^{d,j} : x \in \mathcal{X})$ to denote the vector of expected drifts for resource $j$. We assume that arm $x^0 = 1 \in \mathcal{X}$ is a null arm with three important properties: (i) its reward is zero a.s.; (ii) the drift for each resource is nonnegative a.s.; and (iii) the expected drift for each resource is positive.

---

[1] In fact, our proofs remain valid even if the outcome distribution depends on the past history provided the conditional expectation is independent of the past history and fixed for each arm. In this case $\mu_x^o$ denotes the conditional expectation of $o_t$ when arm $x$ is pulled in round $t$. Since our proofs rely on the Azuma-Hoeffding inequality, we need this assumption on the conditional expectation to hold.

The second and third properties of the null arm plus the model's requirement that $x_t = 1$ if $\exists j$ s.t. $B_{t-1,j} < 1$ ensure that the budgets are nonnegative a.s. and can be safely increased from $0$.

Our model is intended to capture applications featuring resource renewal, such as the following. In each round, each resource gets replenished by some random amount and the chosen arm consumes some random amount of each resource. If the consumption is less than replenishment, the resource gets renewed. The random variable $d_{t,j}$ then models the net replenishment minus consumption. The full model presented above is more general because it allows both the consumption and replenishment to depend on the arm pulled.

We consider two settings in this paper.

**MDP setting** The decision-maker *knows* the true outcome distributions. In this setting the model implicitly defines an MDP, where the state is the budget vector, the actions are arms, and the transition probabilities are defined by the outcome distributions of the arms.

**Learning setting** The decision-maker *does not know* the true outcome distributions.

The goal is to design to an MDP policy for the first setting and a learning algorithm for the second, and bound their regret against an LP relaxation as defined in the next subsection.

## 2.2 Linear Programming Relaxation

Similar to Badanidiyuru et al. [6, Lemma 3.1], we consider the following LP relaxation that provides an upper bound on the total expected reward of any algorithm:

$$\mathsf{OPT}_{\mathsf{LP}} = \max_p \left\{ \sum_{x \in \mathcal{X}} p_x \mu_x^r : \sum_{x \in \mathcal{X}} p_x \mu_x^{d,j} \geq {}^{-B}/_T \ \forall j \in \mathcal{J}, \ \sum_{x \in \mathcal{X}} p_x = 1, \ p_x \geq 0 \ \forall x \in \mathcal{X} \right\}. \quad (1)$$

**Lemma 2.1.** *The total expected reward of any algorithm is at most* $T \cdot \mathsf{OPT}_{\mathsf{LP}}$.

The proof of this lemma, similar to those in existing works [2, 6], follows from the observations that (i) the variables $p = \{p_x : x \in \mathcal{X}\}$ can be interpreted as the probability of choosing arm $x$ in a round; and (ii) if we set $p_x$ equal to the expected number of times $x$ is chosen by an algorithm divided by $T$, then it is a feasible solution for the LP.

**Definition 2.1** (Regret). The regret of an algorithm $\mathcal{A}$ is defined as $R_T(\mathcal{A}) = T \cdot \mathsf{OPT}_{\mathsf{LP}} - \mathsf{REW}(\mathcal{A})$, where $\mathsf{REW}(\mathcal{A})$ denotes the total expected reward of $\mathcal{A}$.

## 2.3 Assumptions

We assume that the initial budget of every resource is $B \leq T$. This assumption is without loss of generality because otherwise we can scale the drifts by dividing them by the smallest budget. This results in a smaller support set for the drift distribution that is still contained in $[-1, 1]$.

Our assumptions about the null arm $x^0$ are a major difference between our model and BwK. In BwK the budgets can only decrease and the process ends when the budget of any resource reaches $0$. However, in our model the budgets can increase or decrease, and the process ends at the end of the time horizon. Our assumptions about the null arm allow us to increase the budget from $0$ without making it negative.[2] A side-effect of this is that in our model we can even assume that $B$ is a small constant because we can always increase the budget by pulling the null arm, in contrast to existing literature on BwK that assume the initial budgets are large and often scale with the time horizon.

A standard assumption for achieving logarithmic regret in stochastic bandits is that the gap between the total expected reward of an optimal arm and that of a second-best arm is positive. There are a few different ways in which one could translate this to our model where the optimal solution is a mixture over arms. We make the following choice. We assume that there exists a unique set of arms $X^*$ that form the support set of the LP solution and a unique set of resources $J^*$ that correspond to binding

---

[2]In a model where, in each round, each resource gets replenished by some random amount and the chosen arm consumes some random amount of each resource, the null arm represents the option to remain idle and do nothing while waiting for resource replenishment. See Appendix A for more discussion on the assumptions about the null arm.

constraints in the LP solution [14]. We define the gap of the problem instance in Definition 4.1 and our uniqueness assumption[3] implies that the gap is strictly positive.

We make a few separation assumptions parameterized by four positive constants that can be arbitrarily small. First, the smallest magnitude of the drifts, $\delta_{\text{drift}} = \min\{|\mu_x^{d,j}| : x \in \mathcal{X}, j \in \mathcal{J}\}$, satisfies $\delta_{\text{drift}} > 0$. Second, the smallest singular value of the LP constraint matrix, denoted by $\sigma_{\min}$, satisfies $0 < \sigma_{\min} < 1$. Third, the LP solution $p^*$ satisfies $p_x^* \geq \delta_{\text{support}} > 0$ for all $x \in X^*$. Fourth, $\sum_{x \in X^*} p_x^* \mu_x^{d,j} \geq \delta_{\text{slack}} > 0$ for all resources $j \notin J^*$. The first assumption is necessary for logarithmic regret bounds because otherwise one can show that the regret of the optimal algorithm for the case of one resoure and one zero-drift arm is $\Theta(\sqrt{T})$ (Appendix B). The second and third assumptions are essentially the same as in existing literature on logarithmic regret bounds for BwK [9, 14]. The fourth assumption allows us to design algorithms that can increase the budgets of the non-binding resources away from $0$, thereby reducing the number of times the algorithm has to pull the null arm. Otherwise, if they have zero drift, then, as stated above, the regret of the optimal algorithm for the case of one resource and one zero-drift arm is $\Theta(\sqrt{T})$ (Appendix B).

## 3 MDP Policy with Constant Regret

In this section we design an MDP policy, `ControlBudget` (Algorithm 2), with constant regret in terms of $T$ for the setting when the learner knows the true outcome distributions and our model implicitly defines an MDP (Section 2.1). At a high level, `ControlBudget`, which shares similarities with Flajolet and Jaillet [9, Algorithm UCB-Simplex], plays arms to keep the budgets close to a decreasing sequence of thresholds. The choice of this sequence allows us to show that the expected leftover budgets and the expected number of null arm pulls are constants. This is a key step in proving the final regret bound. We start by considering the special case of one resource in Section 3.1 because it provides intuition for the general case of multiple resources in Section 3.2.

### 3.1 Special Case: One Resource

Since there is only one resource we drop the superscript $j$ in this section. We say that an arm $x$ is a positive (resp. negative) drift arm if $\mu_x^d > 0$ (resp. $\mu_x^d < 0$). The following lemma characterizes the possible solutions of the LP (Eq. (1)).

**Lemma 3.1.** *The solution of the LP relaxation (Eq. (1)) is supported on at most two arms. Furthermore, if $T \geq {}^B\!/_{\delta_{drift}}$, then the solution belongs to one of three categories: (i) supported on a single positive drift arm; (ii) supported on the null arm and a negative drift arm; (iii) supported on a positive drift arm and a negative drift arm.*

The proof of this lemma follows from properties of LPs and a case analysis of which constraints are tight. Our MDP policy, `ControlBudget` (Algorithm 1), deals with the three cases separately and satisfies the following regret bound.[4]

**Theorem 3.1.** *If $c \geq \frac{6}{\delta_{drift}^2}$, the MDP policy `ControlBudget` (Algorithm 1) satisfies*

$$R_T(\texttt{ControlBudget}) \leq \tilde{C}, \tag{2}$$

*where $\tilde{C} = O\left(\delta_{drift}^{-4} \ln\left(\left(1 - \exp\left(-\frac{\delta_{drift}^2}{8}\right)\right)^{-1}\right) + \delta_{drift}^{-1}\left(1 - \exp\left(\delta_{drift}^2\right)\right)^{-2}\right)$ is a constant.*

We defer all proofs in this section to Appendix C, but we include a proof sketch of most results in the main paper following the statement. The proof of Theorem 3.1 follows from the following sequence of lemmas.

**Lemma 3.2.** *If the LP solution is supported on a positive drift arm $x^p$, then*

$$R_T(\texttt{ControlBudget}) \leq \tilde{C}, \tag{3}$$

---

[3]This assumption is essentially without loss of generality because the set of problem instances with multiple optimal solutions is a set of measure zero.

[4]In this theorem and the rest of the paper, we use $\tilde{C}$ to denote a constant that depends on problem parameters, including $k, m$, and the various separation constants mentioned in Section 2.3, but *does not depend on $T$.* We use this notation because the main focus of this work is how the regret scales as a function of $T$.

---
**Algorithm 1:** `ControlBudget` (for $m = 1$)
---
**Input:** time horizon $T$, initial budget $B$, set of arms $\mathcal{X}$, set of resources $\mathcal{J}$, constant $c > 0$.

1 Set $B_0 = B$.
2 **if** *LP solution is supported on positive drift arm $x^p$* **then**
3      **for** $t = 1, 2, \ldots, T$ **do**
4          If $B_{t-1} < 1$, pull $x^0$. Otherwise, pull $x^p$.
5      **end**
6 **else if** *LP solution is supported on null arm $x^0$ and negative drift arm $x^n$* **then**
7      **for** $t = 1, 2, \ldots, T$ **do**
8          Define threshold $\tau_t = c \log(T - t)$.
9          If $B_{t-1} < \max\{1, \tau_t\}$, pull $x^0$. Otherwise, pull $x^n$.
10      **end**
11 **else if** *LP solution is supported on positive drift arm $x^p$ and negative drift arm $x^n$* **then**
12      **for** $t = 1, 2, \ldots, T$ **do**
13          Define threshold $\tau_t = c \log(T - t)$.
14          If $B_{t-1} < 1$, pull $x^0$. If $1 \le B_{t-1} < \tau_t$, pull $x^p$. Otherwise, pull $x^n$.
15      **end**
---

*where $\tilde{C} = O\left(\delta_{drift}^{-3} \ln\left(\left(1 - \exp\left(-\frac{\delta_{drift}^2}{8}\right)\right)^{-1}\right)\right)$ is a constant.*

We can write the regret in terms of the norm of $\xi = (\xi_{x^p})$, where $\xi_{x^p}$ is the expected difference between the number of times $x^p$ is played by the LP and by `ControlBudget`. This is equal to the expected number of times the policy plays the null arm and, in turn, is equal to the expected number of rounds in which the budget is below 1. Since both $x^0$ and $x^p$ have positive drift, this is a transient random walk that drifts away from 0. It is known that such a walk spends a constant number of rounds in any state in expectation.

**Lemma 3.3.** *If the LP solution is supported on the null arm $x^0$ and a negative drift arm $x^n$, then*

$$R_T(\textit{ControlBudget}) \le \tilde{C} \cdot \mathbb{E}[B_T], \tag{4}$$

*where $\tilde{C} = O(\delta_{drift}^{-1})$ is a constant.*

We can write the regret in terms of the norm of $\xi = (\xi_{x^0}, \xi_{x^n})$, where $\xi_x$ is the expected difference between the number of times $x$ is played by the LP and by `ControlBudget`. Since both constraints (resource and sum-to-one) are tight, the lemma follows by writing $\xi = D^{-1}b$ and taking norms, where $D$ is the LP constraint matrix and $b = (-\mathbb{E}[B_T], 0)$.

**Lemma 3.4.** *If the LP solution is supported on a positive drift arm $x^p$ and a negative drift arm $x^n$, then*

$$R_T(\textit{ControlBudget}) \le \tilde{C} \cdot \max\{\mathbb{E}[B_T], \mathbb{E}[N_{x^0}]\}, \tag{5}$$

*where $\mathbb{E}[N_{x^0}]$ denotes the expected number null arm pulls and $\tilde{C} = O(\delta_{drift}^{-1})$ is a constant.*

This lemma follows similarly to the previous one by writing regret in terms of the norm of $\xi = (\xi_{x^p}, \xi_{x^n})$ and writing $\xi = D^{-1}b$ for $b = (-\mathbb{E}[B_T], \mathbb{E}[N_{x^0}])$.

Therefore, proving that $R_T(\textit{ControlBudget})$ is a constant in $T$ requires proving that both the expected leftover budget and expected number of null arm pulls are constants. Intuitively, we could ensure $\mathbb{E}[B_T]$ is small by playing the negative drift arm whenever the budget is at least 1. However, there is constant probability of the budget decreasing below 1 and the expected number of null arm pulls becomes $O(T)$. `ControlBudget` solves the tension between the two objectives by carefully choosing a decreasing sequence of thresholds $\tau_t$. The threshold is initially far from 0 to ensure low probability of pre-mature resource depletion, but decreases to 0 over time to ensure small expected leftover budget and decreases at a rate that ensures the expected number of null arm pulls is a constant.

**Lemma 3.5.** *If the LP solution is supported on a positive drift arm $x^p$ and a negative drift arm $x^n$, and $c \ge \frac{6}{\delta_{drift}^2}$, then*

$$\mathbb{E}[N_{x^0}] \le \tilde{C}, \tag{6}$$

where $\tilde{C} = O\left(\delta_{drift}^{-3} \ln\left(\left((1 - \exp\left(-\frac{\delta_{drift}^2}{8}\right)\right)^{-1}\right)\right)$ is a constant.

If the budget is below the threshold, i.e., $B_{t-1} < \tau_t$ for some $t$, then `ControlBudget` pulls $x^p$ until $B_s \geq \tau_{s+1}$ for some $s \geq t$. Since $x^p$ has positive drift, the event that repeated pulls decrease the budget towards 0 is a low probability event. Using this, our choice of $\tau_t = c\log(T - t)$ for an appropriate constant $c$, and summing over all rounds shows that the expected number of rounds in which the budget is less than 1 is a constant in $T$.

**Lemma 3.6.** *If the LP solution is supported on two arms, and $c \geq \frac{6}{\delta_{drift}^2}$, then*

$$\mathbb{E}[B_T] \leq \tilde{C}, \tag{7}$$

*where $\tilde{C} = \tilde{O}\left(\left(1 - \exp\left(\delta_{drift}^2\right)\right)^{-2} + \delta_{drift}^{-2}\right)$ is a constant.*

If $B_{t-1} \geq \tau_t$, then `ControlBudget` pulls a negative drift arm $x^n$. We can upper bound the expected leftover budget by conditioning on $q$, the number of consecutive pulls of $x^n$ at the end of the timeline. The main idea in completing the proof is that (i) if $q$ is large, then it corresponds to a low probability event; and (ii) if $q$ is small, then the budget in round $T - q$ was smaller than $\tau_q$, which is a decreasing sequence in $q$, and there are few rounds left so the budget cannot increase by too much.

## 3.2 General Case: Multiple Resources

Now we use the ideas from Section 3.1 to tackle the case of $m > 1$ resources that is much more challenging. Generalizing Lemma 3.1, the solution of the LP relaxation (Eq. (1)) is supported on at most $\min\{k, m\}$ arms. Informally, our MDP policy, `ControlBudget` (Algorithm 2), samples an arm from a probability distribution that ensures drifts bounded away from 0 in the "correct directions": (i) a binding resource $j$ has drift at least $\gamma_t$ if $B_{t-1,j} < \tau_t$ and drift at most $-\gamma_t$ if $B_{t-1,j} \geq \tau_t$; and (ii) a non-binding resource $j$ has drift at least $\frac{1}{2}\gamma_t$ if $B_{t-1,j} < \tau_t$. This allows us to show that the expected leftover budget for each binding resource and the expected number of null arm pulls are constants in terms of $T$.

---

**Algorithm 2:** `ControlBudget` (for general $m$)

---

**Input:** time horizon $T$, initial budget $B$, set of arms $\mathcal{X}$, set of resources $\mathcal{J}$, constant $c > 0$.

1 Set $B_{0,j} = B$ for all $j \in \mathcal{J}$.
2 Define threshold $\tau_t = c\log(T - t)$.
3 **for** $t = 1, 2, \ldots, T$ **do**
4     **if** $\exists j \in \mathcal{J}$ *such that* $B_{t-1,j} < 1$ **then**
5         Pull the null arm $x^0$.
6     **else**
7         Define $s_t \in \{\pm 1\}^{|X^*|-1} \times 0$ as follows. Let $j$ denote the resource corresponding to row $i \in [|X^*| - 1]$ in the matrix $D$ and vector $b$. Then, the $i$th entry of $s_t$ is $+1$ if $B_{t-1,j} < \tau_t$ and $-1$ otherwise.
8         Define $\gamma_t$ to be the solution to the following constrained optimization problem:

$$\max_{\gamma \in [0,1]} \left\{\gamma : p = D^{-1}(b + \gamma s_t) \geq 0, \ p^T \mu^{d,j} \geq \frac{\gamma}{2} \ \forall j \in \mathcal{J} \setminus J^* \text{ if } B_{t-1,j} < \tau_t\right\}. \tag{8}$$

9         Sample an arm from the probability distribution $p_t = D^{-1}(b + \gamma_t s_t)$.
10     **end**
11 **end**

---

**Theorem 3.2.** *If $c \geq \frac{6}{\gamma^{*2}}$, the regret of `ControlBudget` (Algorithm 2) satisfies*

$$R_T(\textit{ControlBudget}) \leq \tilde{C}, \tag{9}$$

*where $\gamma^*$ (defined in Lemma 3.9) and $\tilde{C}$ are constants with $\tilde{C} = O\left(m\sigma_{\min}^{-1}\left(m(\gamma^*)^{-3} \ln\left(\left((1 - \exp\left(-\gamma^{*2}\right)\right)^{-1}\right) + \left(1 - \exp(\gamma^{*2})\right)^{-2}\right)\right)$.*

We defer all proofs in this section to Appendix D, but we include a proof sketch of most results in the main paper following the statement. The proof of Theorem 3.2 follows from the following sequence of lemmas. The next two lemmas are generalizations of Lemmas 3.3 and 3.4 with essentially the same proofs. Recall $J^*$ denotes the unique set of resources that correspond to binding constraints in the LP solution (Section 2.3).

**Lemma 3.7.** *If the LP solution includes the null arm $x^0$ in its support, then*

$$R_T(\texttt{ControlBudget}) \leq \tilde{C} \cdot \left( \sum_{j \in J^*} \mathbb{E}[B_{T,j}] \right), \tag{10}$$

*where $\tilde{C} = O(\sigma_{\min}^{-1})$ is a constant.*

**Lemma 3.8.** *If the LP solution does not include the null arm $x^0$ in its support, then*

$$R_T(\texttt{ControlBudget}) \leq \tilde{C} \cdot \left( \sum_{j \in J^*} \mathbb{E}[B_{T,j}] + \mathbb{E}[N_{x^0}] \right), \tag{11}$$

*where $\mathbb{E}[N_{x^0}]$ denotes the expected number of null arm pulls and $\tilde{C} = O(\sigma_{\min}^{-1})$ is a constant.*

Lemmas 3.10 and 3.11 are generalizations of Lemmas 3.5 and 3.6 with similar proofs after taking a union bound over resources. But we first need Lemma 3.9 that lets us conclude there is drift of magnitude at least $\gamma^* > 0$ in the "correct directions" as stated earlier.

**Lemma 3.9.** *[9, Lemma 14] In each round $t$, $\gamma_t \geq \gamma^* = \frac{\sigma_{\min} \min\{\delta_{support}, \delta_{slack}\}}{4m}$.*

The proof of this lemma is identical to Flajolet and Jaillet [9, Lemma 14] but we provide a proof in the appendix for completeness.

**Lemma 3.10.** *If the LP solution does not include the null arm in its support, then*

$$\mathbb{E}[N_{x^0}] \leq \tilde{C}, \tag{12}$$

*where $\tilde{C} = O\left( m(\gamma^*)^{-3} \ln \left( \left( \left(1 - \exp\left(-\frac{\gamma^{*2}}{8}\right)\right)^{-1} \right) \right) \right)$ is a constant.*

**Lemma 3.11.** *If the LP solution is supported on more than one arm, then for all $j \in J^*$*

$$\mathbb{E}[B_{T,j}] \leq \tilde{C}, \tag{13}$$

*where $\tilde{C} = \tilde{O}\left( \left(1 - \exp(\gamma^{*2})\right)^{-2} + (\gamma^*)^{-2} \right)$ is a constant.*

A subtle but important point is that the regret analysis does not require `ControlBudget` to know the true expected drifts in order to find the probability vector $p_t$. It simply requires the algorithm to know $X^*$, $J^*$, and find any probability vector $p_t$ that ensures drifts bounded away from 0 in the "correct directions" as stated earlier. We use this property in our learning algorithm, `ExploreThenControlBudget` (Algorithm 3), in the next section.

## 4 Learning Algorithm with Logarithmic Regret

In this section we design a learning algorithm, `ExploreThenControlBudget` (Algorithm 3), with logarithmic regret in terms of $T$ for the setting when the learning does not know the true distributions. Our algorithm, which can be viewed as combining aspects of Li et al. [14, Algorithm 1] and Flajolet and Jaillet [9, Algorithm UCB-Simplex], proceeds in three phases. It uses phase one of Li et al. [14, Algorithm 1] to identify the set of optimal arms $X^*$ and the set of binding constraints $J^*$ by playing arms in a round-robin fashion, and using confidence intervals and properties of LPs. This is reminiscent of successive elimination [8], except that the algorithm tries to identify the optimal arms instead of eliminating suboptimal ones. In the second phase the algorithm continues playing the arms in $X^*$ in a round-robin fashion to shrink the confidence radius further. In the third phase the algorithm plays a variant of the MDP policy `ControlBudget` (Algorithm 2) with a slighly different optimization problem for $\gamma_t$ because it only has empirical esitmates of the drifts.

## 4.1 Additional Notation and Preliminaries

For all arms $x \in \mathcal{X}$ and rounds $t \geq k$, define the upper confidence bound (UCB) of the expected outcome vector $\mu_x^o$ as $\mathsf{UCB}_t(x) = \bar{o}_t(x) + \mathsf{rad}_t(x)$, where $\mathsf{rad}_t(x) = \sqrt{8n_t(x)^{-1} \log T}$ denotes the confidence radius, $n_t(x)$ denotes the number of times $x$ has been played before $t$, and $\bar{o}_t(x) = n_t(x)^{-1} \sum_t o_t \mathbb{1}[x_t = x]$ denotes the empirical mean outcome vector of $x$. The lower confidence bound (LCB) is defined similarly as $\mathsf{LCB}_t(x) = \bar{o}_t(x) - \mathsf{rad}_t(x)$.

For all arms $x \in \mathcal{X}$, let $\mathsf{OPT}_{-x}$ denote the value of the LP relaxation (Eq. (1)) with the additional constraint $p_x = 0$, and for all resources $j \in \mathcal{J}$, let $\mathsf{OPT}_{-j}$ denote the value when the objective has an extra $-\sum_x p_x \mu_x^{d,j} + B/T$ term [14]. Intuitively, these represent how important it is to play arm $x$ or make the resource constraint for $j$ a binding constraint. Define the UCB of $\mathsf{OPT}_{-x}$ to be the value of the LP when the expected outcome is replaced by its UCB, and denote this by $\mathsf{UCB}_t(\mathsf{OPT}_{-x})$. The LCB for $\mathsf{OPT}_{-x}$, and UCB and LCB for $\mathsf{OPT}_{-j}$ and $\mathsf{OPT}_{\mathsf{LP}}$ are defined similarly.

**Definition 4.1** (Gap [14]). The gap of the problem instance is defined as

$$\Delta = \min \left\{ \min_{x \in X^*} \{\mathsf{OPT}_{\mathsf{LP}} - \mathsf{OPT}_{-x}\}, \min_{j \notin J^*} \{\mathsf{OPT}_{\mathsf{LP}} - \mathsf{OPT}_{-j}\} \right\}. \tag{14}$$

## 4.2 Learning Algorithm and Regret Analysis

---

**Algorithm 3:** `ExploreThenControlBudget`

---

**Input:** time horizon $T$, initial budget $B$, set of arms $\mathcal{X}$, set of resources $\mathcal{J}$, constant $c > 0$.

1   Set $B_{0,j} = B$ for all $j \in \mathcal{J}$.
2   Initialize $t = 1, X^* = \emptyset, J' = \emptyset$.
3   **while** $t < T - k$ and $|X^*| + |J'| < m + 1$ **do**
4     Play each arm in $\mathcal{X} \setminus \{x^0\}$ in a round-robin fashion. Play $x^0$ if $\exists j$ such that $B_{t-1,j} < 1$.
5     For each $x \in \mathcal{X}$, if $\mathsf{UCB}_t(\mathsf{OPT}_{-x}) < \mathsf{LCB}_t(\mathsf{OPT}_{\mathsf{LP}})$, then add $x$ to $X^*$.
6     For each $j \in \mathcal{J}$, if $\mathsf{UCB}_t(\mathsf{OPT}_{-j}) < \mathsf{LCB}_t(\mathsf{OPT}_{\mathsf{LP}})$, then add $j$ to $J'$.
7   **end**
8   Set $J^* = \mathcal{J} \setminus J'$.
9   **while** $t < T - |X^*|$ and $n_t(x) < \frac{32 \log T}{\gamma^{*2}}$ for all $x \in X^*$, where $\gamma^*$ is defined in Lemma 3.9 **do**
10    Play each arm in $X^*$ in a round-robin fashion.
11   **end**
12   **while** $t < T$ **do**
13    **if** $\exists j \in \mathcal{J}$ such that $B_{t-1,j} < 1$ **then**
14      Pull the null arm $x^0$.
15    **else**
16      Define $s_t$ as in Algorithm 2.
17      Choose $(\gamma_t, p_t) = \max_{\gamma \in [0,1]} \gamma$ such that there exists a probability vector $p$ satisfying

$$\mathsf{LCB}_t(p^T \mu^{d,j}) \geq \frac{\gamma}{8} \ \forall j \in \mathcal{J} \setminus J^* \text{ if } B_{t-1,j} < \tau_t, \tag{15}$$

$$\mathsf{LCB}_t(p^T \mu^{d,j}) \geq \frac{\gamma}{8} \ \forall j \in J^* \text{ if } B_{t-1,j} < \tau_t \tag{16}$$

$$\mathsf{UCB}_t(p^T \mu^{d,j}) \leq -\frac{\gamma}{8} \ \forall j \in J^* \text{ if } B_{t-1,j} \geq \tau_t. \tag{17}$$

18      Sample an arm from the probability distribution $p_t$.
19    **end**
20   **end**

---

**Theorem 4.1.** *If $c \geq \frac{6}{\gamma^*}$, the regret of* `ExploreThenControlBudget` *(Algorithm 3) satisfies*

$$R_T(\texttt{ExploreThenControlBudget}) \leq \tilde{C} \cdot \log T, \tag{18}$$

where $\gamma^*$ (defined in Lemma 3.9), $\tilde{C}'$ and $\tilde{C}$ are constants with $\tilde{C}'$ denoting the constant in Theorem 3.2 and

$$\tilde{C} = O\left(\frac{km^2}{\min\{\delta_{drift}^2, \sigma_{\min}^2\}\Delta^2} + k(\gamma^*)^{-2} + \tilde{C}'\right). \tag{19}$$

We defer all proofs in this section to Appendix E, but we include a proof sketch of most results in the main paper following the statement. We refer to the three while loops of `ExploreThenControlBudget` (Algorithm 3) as the three phases. The lemmas and corollaries following the theorem below show that the first phase consists of at most logarithmic number of rounds. It is easy to see that the second phase consists of at most logarithmic number of rounds. The third phase plays a variant of the MDP policy `ControlBudget` (Algorithm 2). There exists a feasible solution to the optimization problem in Line 17 that ensures drifts bounded away from 0 in the "correct directions" (Appendix E.3). Combining the analysis of the first two phases with Theorem 3.2 lets us conclude that `ExploreThenControlBudget` has logarithmic regret.

**Definition 4.2** (Clean Event). The clean event is the event such that for all $x \in \mathcal{X}$ and all $t \geq k$, (i) $\mu_x^0 \in [\mathsf{LCB}_t(x), \mathsf{UCB}_t(x)]$; and (ii) after the first $n$ pulls of the null arm the sum of the drifts for each resource is at least $w$, where $w = \frac{4096km^2 \log T}{\delta_{drift}^2 \Delta^2}$ and $n = \frac{2w}{\mu_{x^0}^d}$.

**Lemma 4.1.** *The clean event occurs with probability at least* $1 - 5mT^{-2}$.

The lemma follows from the Azuma-Hoeffding inequality. Since the complement of the clean event contributes $O(mT^{-1})$ to the regret, it suffices to bound the regret conditioned on the clean event.

**Lemma 4.2.** *If the clean event occurs, then* $\mathsf{UCB}_t(\mathsf{OPT_{LP}}) - \mathsf{LCB}_t(\mathsf{OPT_{LP}}) \leq \frac{8m}{\sigma_{\min}}\mathsf{rad}_t$. *A similar statement is true for* $\mathsf{OPT}_{-x}$ *and* $\mathsf{OPT}_{-j}$ *for all* $x \in \mathcal{X}$ *and* $j \in \mathcal{J}$.

The proof follows from a perturbation analysis of the LP and uses the confidence radius to bound the perturbations in the rewards and drifts.

**Corollary 4.1.** *If the clean event occurs and* $n_t(x) > \frac{2048m^2 \log T}{\sigma_{\min}^2 \Delta^2}$ *for all* $x \in \mathcal{X}$, *then*

$$\mathsf{UCB}_t(\mathsf{OPT}_{-x}) < \mathsf{LCB}_t(\mathsf{OPT_{LP}}) \text{ and } \mathsf{UCB}_t(\mathsf{OPT}_{-j}) < \mathsf{LCB}_t(\mathsf{OPT_{LP}}) \tag{20}$$

*for all* $x \in X^*$ *and* $j \in \mathcal{J} \setminus J^*$.

This follows from substituting the bound on $n_t(x)$ into the definition of $\mathsf{rad}_t(x)$ and applying Lemma 4.2. In the worst case, each pull of an arm can cause the budget to drop below 1, but the clean event implies that the first $n$ pulls of $x^0$ have enough total drift to allow $n_t(x)$ pulls of each non-null arm in phase 1. This allows us to upper bound the duration of phase 1 as follows.

**Corollary 4.2.** *If the clean event occurs, then phase 1 of* `ExploreThenControlBudget` *has at most*

$$\tilde{C} \cdot \log T \tag{21}$$

*rounds, where* $\tilde{C} = O\left(\frac{km^2}{\min\{\delta_{drift}^2, \sigma_{\min}^2\}\Delta^2}\right)$ *is a constant.*

### 4.3 Reduction from BwK

**Theorem 4.2.** *Suppose* $\frac{B}{T} \geq \delta_{drift}$. *Consider a* BwK *instance such that (i) for each arm and resource, the expected consumption of that resource differs from* $\frac{B}{T}$ *by at least* $\delta_{drift}$; *and (ii) all the other assumptions required by Theorem 4.1 (Section 2.3) are also satisfied. Then, there is an algorithm for* BwK *whose regret satisfies the same bound as in Theorem 4.1 with the same constant* $\tilde{C}$.

Due to space constraints, we present the reduction in Appendix E.4.

## 5 Conclusion

In this paper we introduced a natural generalization of BwK that allows non-monotonic resource utilization. We first considered the setting when the decision-maker knows the true distributions and presented an MDP policy with *constant* regret against an LP relaxation. Then we considered

the setting when the decision-maker does not know the true distributions and presented a learning algorithm with *logarithmic* regret against the same LP relaxation. Finally, we also presented a reduction from BwK to our model and showed a regret bound that matches existing results [14].

An important direction for future research is to obtain optimal regret bounds. The regret bound for our algorithm scales as $O(\text{poly}(k)\text{poly}(m)\Delta^{-2})$, where $k$ is the number of arms, $m$ is the number of resources and $\Delta$ is the suboptimality parameter. A modification to our algorithm along the lines of Flajolet and Jaillet [9, algorithm UCB-Simplex] that considers each support set of the LP solution explicitly leads to a regret bound that scales as $O(\text{poly}(k)2^m\Delta^{-1})$. It is an open question, even for BwK, to obtain a regret bound that scales as $O(\text{poly}(k)\text{poly}(m)\Delta^{-1})$ or show that the trade-off between the dependence on the number of resources and the suboptimality parameter is unavoidable.

Another natural follow-up to our work is to develop further extensions, such as considering an infinte set of arms [11], studying adversarial observations [4, 10], or incorporating contextual information [16, 1] as has been the case elsewhere throughout the literature on bandits.

## Acknowledgments and Disclosure of Funding

We thank Kate Donahue, Sloan Nietert, Ayush Sekhari, and Alex Slivkins for helpful discussions. This research was partially supported by the NSERC Postgraduate Scholarships-Doctoral Fellowship 545847-2020.

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
