# Contents

# A  Assumptions about the Null Arm

In any model in which resources can be consumed and/or replenished over time, one must specify what happens when the budget of one (or more) resources reaches zero. The original bandits with knapsacks problem assumes than when this happens, the process of learning and gaining rewards ceases. The key distinction between that model and ours is that we instead assume the learner is allowed remain idle until the supply of every resource becomes positive again, at which point the learning process recommences. The null arm in our paper is intended to represent this option to remain idle and wait for resource replenishment. In order for these idle periods to have finite length almost surely, a minimal assumption is that when the null arm is pulled, for each resource there is a positive probability that the supply of the resource increases. We make the stronger assumption that for each resource, the expected change in supply is positive when the null arm is pulled. In fact, our results for the MDP setting hold under the following more general assumption: there exists a probability distribution over arms, such that when a random arm is sampled from this distribution and pulled, the expected change in the supply of each resource is positive. In the following, we refer to this as Assumption PD (for "positive drift").

To see that our results for the MDP setting continue to hold under Assumption PD (i.e., even if one doesn't assume that the null arm itself is guaranteed to yield positive expected drift for each resource) simply modify Algorithms 2 and 3 so that whenever they pull the null arm in a time step when the supply of each resource is at least 1, the modified algorithms instead pull a random arm sampled from the probability distribution over arms that guarantees positive expected drift for every resource. As long as the constant $\delta_{\text{drift}}$ is less than or equal to this positive expected drift, the modification to the algorithms does not change their analysis. We believe it's likely that our learning algorithm (Algorithm 3) could similarly be adapted to work under Assumption PD, but it would be less straightforward because the positive-drift distribution over arms would need to be learned.

When Assumption PD is violated, the problem becomes much more similar to the Bandits with Knapsacks problem. To see why, consider a two-player zero-sum game in which the row player chooses an arm $x$, the column player chooses a resource $j$, and the payoff is the expected drift of that resource when that arm is pulled, $\mu_x^{d,j}$. Assumption PD is equivalent to the assertion that the value of the game is positive; the negation of Assumption PD means that the value of the game is negative. By the Minimax Theorem, this means there is a convex combination of resources (i.e., a mixed strategy for the column player) such that the weighted-average supply of these resources is guaranteed to experience non-positive expected drift, no matter which arm is pulled. Either the expected drift is zero — we prove in Appendix A of the supplementary material that regret $O(\sqrt{T})$ is unavoidable in this case — or the expected drift is strictly negative, in which case the weighted-average resource supply inevitably dwindles to zero no matter which arms the learner pulls. In either case, the behavior of the model is qualitatively different when Assumption PD does not hold.

# B  Regret Bounds for One Arm, One Resource, and Zero Drift

In this section we will consider the case when $\mathcal{X} = \{x^0, x\}$, $\mathcal{J} = \{1\}$, and $x$ has zero drift, i.e., $\mu_x^d = 0$. Since $x$ is the only arm besides the null arm, we assume without loss of generality that its reward is equal to 1 deterministically. The optimal policy is to pull $x^0$ when $B_{t-1} < 1$ and $x$ otherwise. We will show that the regret of this policy is $\Theta(\sqrt{T})$.

**Theorem B.1.** *The regret of the MDP policy is $O(\sqrt{T})$.*

*Proof.* The optimal solution of the LP relaxation (Eq. (1)) is $p_x = 1$ and $p_{x^0} = 0$. Since $x^0$ and $x$ have reward equal to $0$ and $1$ deterministically, $\mathsf{OPT}_{\mathsf{LP}} = 1$. Therefore, the regret of the MDP policy is equal to the expected number rounds in which the budget is less than 1. That is,

$$R_T = \mathbb{E}\left[\sum_{t=1}^{T} \mathbb{1}[B_{t-1} < 1]\right].$$

Since $\mathbb{E}[d_t] = 0$ when $B_{t-1} \geq 1$ and $\mathbb{E}[d_t] = \mu_{x^0}^d$ when $B_{t-1} < 1$, we can write

$$
\begin{aligned}
R_T &= \mathbb{E}\left[\sum_{t=1}^{T} \mathbb{1}[B_{t-1} < 1]\right] \\
&= \frac{1}{\mu_{x^0}^d} \mu_{x^0}^d \mathbb{E}\left[\sum_{t=1}^{T} \mathbb{1}[B_{t-1} < 1]\right] \\
&= \frac{1}{\mu_{x^0}^d} \mathbb{E}\left[\sum_{t=1}^{T} \mu_{x^0}^d \mathbb{1}[B_{t-1} < 1] + 0\mathbb{1}[B_{t-1} \geq 1]\right] \\
&= \frac{1}{\mu_{x^0}^d} \mathbb{E}\left[\sum_{t=1}^{T} d_t\right] \\
&= \frac{1}{\mu_{x^0}^d} \left(\mathbb{E}[B_T] - B\right).
\end{aligned}
$$

Since $B_0 = B$, the budget is updated as $B_t = B_{t-1} + d_t$ and $d_t \in [-1, 1]$, we have

$$
\begin{aligned}
\mathbb{E}\left[B_t^2 | B_{t-1}\right] &= \mathbb{E}\left[B_{t-1}^2 + 2B_{t-1}d_t + d_t^2 | B_{t-1}\right] \\
&= \mathbb{E}\left[B_{t-1}^2 | B_{t-1}\right] + \mathbb{E}\left[2B_{t-1}d_t | B_{t-1}\right] + \mathbb{E}\left[d_t^2 | B_{t-1}\right] \\
&\leq B_{t-1}^2 + \mathbb{E}\left[2B_{t-1}d_t | B_{t-1}\right] + 1^2 \\
&= B_{t-1}^2 + 2B_{t-1}\mu_{x^0}^d \mathbb{1}[B_{t-1} < 1] + 1 \\
&\leq B_{t-1}^2 + 2\mu_{x^0}^d + 1 \\
\Rightarrow \mathbb{E}\left[B_T^2\right] &= O(T).
\end{aligned}
$$

Using Jensen's inequality, we have

$$
\mathbb{E}[B_T] \leq \sqrt{\mathbb{E}[B_T^2]} = O(\sqrt{T}).
$$

This completes the proof. $\qquad\qquad\qquad\qquad\qquad\qquad\qquad\qquad\qquad\qquad\qquad\square$

**Theorem B.2.** *If $\mathbb{E}\left[d_t^2 | B_{t-1}\right] \geq \sigma^2 > 0$, then the regret of the MDP policy is $\Omega(\sqrt{T})$.*

*Proof.* Using the proof of Theorem B.1, it suffices to provide a lower bound on $\mathbb{E}[B_T]$. Since the budget is updated as $B_t = B_{t-1} + d_t$, $\mathbb{E}[d_t | B_{t-1}] \geq 0$, and $\mathbb{E}\left[d_t^2 | B_{t-1}\right] \geq \sigma^2$, we have

$$
\begin{aligned}
\mathbb{E}\left[B_t^2 | B_{t-1}\right] &= \mathbb{E}\left[B_{t-1}^2 + 2B_{t-1}d_t + d_t^2 | B_{t-1}\right] \\
&= \mathbb{E}\left[B_{t-1}^2 | B_{t-1}\right] + \mathbb{E}\left[2B_{t-1}d_t | B_{t-1}\right] + \mathbb{E}\left[d_t^2 | B_{t-1}\right] \\
&\geq B_{t-1}^2 + \mathbb{E}\left[2B_{t-1}d_t | B_{t-1}\right] + \sigma^2 \\
&= B_{t-1}^2 + 2B_{t-1}\mathbb{E}[d_t | B_{t-1}] + \sigma^2 \\
&\geq B_{t-1}^2 + \sigma^2 \\
\Rightarrow \mathbb{E}\left[B_T^2\right] &\geq \Omega(T).
\end{aligned}
$$

The Cauchy-Schwarz inequality yields that

$$
\mathbb{E}\left[\left(B_T^{1/2}\right)^2\right]^{1/2} \mathbb{E}\left[\left(B_T^{3/2}\right)^2\right]^{1/2} \geq \mathbb{E}\left[B_T^2\right] \geq \Omega(T).
$$

Squaring both sides yields that

$$
\mathbb{E}[B_T] \mathbb{E}\left[B_T^3\right] \geq \mathbb{E}\left[B_T^2\right]^2 \geq \Omega(T^2).
$$

It suffices to show that $\mathbb{E}\left[B_T^3\right] = O(T^{3/2})$ because this will imply that $\mathbb{E}\left[B_T\right] = \Omega(T^{1/2})$. Since $d_t \in [-1, 1]$, we have

$$
\begin{aligned}
&\mathbb{E}\left[B_t^3 | B_{t-1}\right] \\
&= \mathbb{E}\left[B_{t-1}^3 + 3B_{t-1}^2 d_t + 3B_{t-1}d_t^2 + d_t^3 | B_{t-1}\right] \\
&= B_{t-1}^3 + 3B_{t-1}^2 \mathbb{E}\left[d_t | B_{t-1}\right] + 3B_{t-1}\mathbb{E}\left[d_t^2 | B_{t-1}\right] + \mathbb{E}\left[d_t^3 | B_{t-1}\right] \\
&= B_{t-1}^3 + 3B_{t-1}^2 \mu_{x^0}^d \mathbb{1}[B_{t-1} < 1] + 3B_{t-1}\mathbb{E}\left[d_t^2 | B_{t-1}\right] + \mathbb{E}\left[d_t^3 | B_{t-1}\right] \\
&\leq B_{t-1}^3 + 3\mu_{x^0}^d + 3B_{t-1} + 1.
\end{aligned}
$$

Taking expectation on both sides yields

$$
\begin{aligned}
\mathbb{E}\left[B_t^3\right] &\leq \mathbb{E}\left[B_{t-1}^3\right] + 3\mathbb{E}\left[B_{t-1}\right] + O(1) \\
&\leq \mathbb{E}\left[B_{t-1}^3\right] + O(\sqrt{t}),
\end{aligned}
$$

where the last inequality follows from the proof of Theorem B.1 where we show that $\mathbb{E}\left[B_t\right] \leq O(\sqrt{t})$. Summing both sides over all rounds yields that

$$
\mathbb{E}\left[B_T^3\right] = O(T^{3/2})
$$

and this completes the proof. $\qquad\square$

## C Proofs for Section 3.1

### C.1 Proof of Lemma 3.2

When the LP solution is supported on a positive drift arm $x^p$, $\mathsf{OPT}_{\mathsf{LP}} = 1$ because the LP plays it with probability 1. Therefore, the regret is equal to the expected number of times `ControlBudget` (Algorithm 1) pulls the null arm. This, in turn, is equal to the expected number of rounds in which the budget is less than 1.

Define

$$
b_0 = 8\delta_{\mathrm{drift}}^{-2} \ln\left(\frac{2}{1 - \exp\left(-\frac{\delta_{\mathrm{drift}}^2}{8}\right)}\right). \tag{22}
$$

Then, we have that for all $b \geq b_0$,

$$
\sum_{k=b}^{\infty} \mathbf{Pr}\left[B_{s+k} \in [0, 1) | B_s = b\right] \leq \sum_{k=b}^{\infty} \exp\left(-\frac{\delta_{\mathrm{drift}}^2 k}{8}\right) \tag{23}
$$

$$
= \exp\left(-\frac{\delta_{\mathrm{drift}}^2 b}{8}\right)\left(1 - \exp\left(-\frac{\delta_{\mathrm{drift}}^2}{8}\right)\right)^{-1}. \tag{24}
$$

where the first inequality follows from Azuma-Hoeffding's inequality. By our choice of $b_0$, we have that

$$
\sum_{k=b}^{\infty} \mathbf{Pr}\left[B_{s+k} \in [0, 1) | B_s = b\right] \leq \frac{1}{2}. \tag{25}
$$

In words, the probability that the budget ever drops below 1 once it exceeds $b_0$ is at most $\frac{1}{2}$. Now, consider the following recursive definition for two disjoint sequence of indices $s_i$ and $s_i'$. Let $s_0 = \min\{t \geq 1 : B_{t-1} \in [0, 1)\}$, and define

$$
s_i' = \min\{t > s_i : B_{t-1} \geq b_0 \text{ or } t - 1 = T\} \tag{26}
$$

$$
s_{i+1} = \min\{t > s_i' : B_{t-1} \in [0, 1)\}. \tag{27}
$$

In words, $s_i'$ denotes the first round after $s_i$ in which the budget is at least $b_0$ and $s_{i+1}$ denotes the first round after $s_i'$ in which the budget is less than 1. Note that Eq. (25) implies that

$$
\mathbf{Pr}\left[s_i \text{ is defined } | s_{i-1}' \text{ is defined}\right] \leq \frac{1}{2}. \tag{28}
$$

Therefore,

$$\mathbf{Pr}\left[s_i \text{ is defined}\right] \leq \prod_{j=1}^{i} \mathbf{Pr}\left[s_j \text{ is defined} \mid s'_{j-1} \text{ is defined}\right] \leq \frac{1}{2^i}. \tag{29}$$

Now, we can upper bound the expected number of rounds in which the budget is below 1 as

$$\mathbb{E}\left[\sum_{t=1}^{T} \mathbb{1}[B_{t-1} < 1]\right] = \sum_{i=0}^{T-1} \mathbf{Pr}\left[s_i \text{ is defined}\right] \mathbb{E}\left[\sum_{t=s_i}^{s'_i} \mathbb{1}[B_{t-1} < 1]\right] \tag{30}$$

$$\leq \sum_{t=0}^{T-1} 2^{-i} \mathbb{E}\left[\sum_{t=s_i}^{s'_i} \mathbb{1}[B_{t-1} < 1]\right] \tag{31}$$

$$\leq \sum_{t=0}^{T-1} 2^{-i} \mathbb{E}\left[s'_i - s_i\right] \tag{32}$$

$$\leq \sum_{t=0}^{T-1} 2^{-i} \frac{1}{\delta_{\text{drift}}} \mathbb{E}\left[B_{s'_i} - B_{s_i}\right] \tag{33}$$

$$\leq \sum_{t=0}^{T-1} 2^{-i} \frac{1}{\delta_{\text{drift}}} (b_0 + 1) \tag{34}$$

$$\leq 2 \frac{b_0 + 1}{\delta_{\text{drift}}}, \tag{35}$$

where Eq. (33) follows because both the null arm and the positive drift arm have drift at least $\delta_{\text{drift}}$. Therefore, we have that

$$\mathbb{E}\left[\sum_{t=1}^{T} \mathbb{1}[B_{t-1} < 1]\right] \leq \tilde{C}, \tag{36}$$

where

$$\tilde{C} = O\left(\delta_{\text{drift}}^{-3} \ln\left(\frac{2}{1 - \exp\left(-\frac{\delta_{\text{drift}}^2}{8}\right)}\right)\right). \tag{37}$$

### C.2 Proof of Lemma 3.3

Let $p^*$ denote the optimal solution to the LP relaxation and note that $Tp_x^*$ denotes the expected number of times the LP plays arm $x$. Since the LP solution is supported on two arms, both the budget and sum-to-one constraints are tight. Therefore, we have

$$D(Tp^*) = b_{\text{LP}}, \tag{38}$$

where

$$D = \begin{bmatrix} \mu_{x^0}^d & \mu_{x^n}^d \\ 1 & 1 \end{bmatrix}, \ p^* = \begin{bmatrix} p_{x^0}^* \\ p_{x^n}^* \end{bmatrix}, \ b_{\text{LP}} = \begin{bmatrix} -B \\ T \end{bmatrix}. \tag{39}$$

Let $N_x$ denote the number of times ControlBudget (Algorithm 1) plays arm $x$. Since it plays the null arm $x^0$ and the negative drift arm $x^n$, the sum-to-one constraint is tight. However, the budget constraint may not be tight because there may be leftover budget. Therefore, we have

$$DN = b_{\text{LP}} - b, \tag{40}$$

where

$$N = \begin{bmatrix} \mathbb{E}[N_{x^0}] \\ \mathbb{E}[N_{x^n}] \end{bmatrix}, \ b = \begin{bmatrix} -E[B_T] \\ 0 \end{bmatrix}. \tag{41}$$

Define

$$\xi = \begin{bmatrix} \xi_{x^0} \\ \xi_{x^n} \end{bmatrix} = \begin{bmatrix} Tp_{x^0}^* - \mathbb{E}[N_{x^0}] \\ Tp_{x^n}^* - \mathbb{E}[N_{x^n}] \end{bmatrix}. \tag{42}$$

Subtracting Eq. (40) from Eq. (38) we have $\xi = D^{-1}b$, where the LP constraint matrix $D$ is invertible by our assumption that the drifts are nonzero. Finally, letting $\mu^r$ denote the vector of expected rewards, the regret can be expressed as

$$R_T(\texttt{ControlBudget}) = \xi^T \mu^r \tag{43}$$

$$\leq |\xi^T \mu^r| \tag{44}$$

$$\leq \|\xi\|_1 \|\mu^r\|_\infty \tag{45}$$

$$\leq \|D^{-1}\|_1 \|b\|_1 \tag{46}$$

$$\leq C_{\delta_{\text{drift}}} \mathbb{E}[B_T], \tag{47}$$

where $C_{\delta_{\text{drift}}} = O(\delta_{\text{drift}}^{-1})$ is a constant. This completes the proof.

### C.3 Proof of Lemma 3.4

Let $p^*$ denote the optimal solution to the LP relaxation and note that $Tp_x^*$ denotes the expected number of times the LP plays arm $x$. Since the LP solution is supported on two arms, both the budget and sum-to-one constraints are tight. Therefore, we have

$$D(Tp^*) = b_{\text{LP}}, \tag{48}$$

where

$$D = \begin{bmatrix} \mu_{x^p}^d & \mu_{x^n}^d \\ 1 & 1 \end{bmatrix}, \; p^* = \begin{bmatrix} p_{x^p}^* \\ p_{x^n}^* \end{bmatrix}, \; b_{\text{LP}} = \begin{bmatrix} -B \\ T \end{bmatrix}. \tag{49}$$

Let $N_x$ denote the number of times $\texttt{ControlBudget}$ (Algorithm 1) plays arm $x$. Since it plays the null arm $x^0$ when the budget is less than 1 and may have leftover budget, neither the budget nor the sum-to-one constraint are tight. Therefore, we have

$$DN = b_{\text{LP}} - b, \tag{50}$$

where

$$N = \begin{bmatrix} \mathbb{E}[N_{x^p}] \\ \mathbb{E}[N_{x^n}] \end{bmatrix}, \; b = \begin{bmatrix} -E[B_T] \\ \mathbb{E}[N_{x^0}] \end{bmatrix}. \tag{51}$$

Define

$$\xi = \begin{bmatrix} \xi_{x^p} \\ \xi_{x^n} \end{bmatrix} = \begin{bmatrix} Tp_{x^p}^* - \mathbb{E}[N_{x^p}] \\ Tp_{x^n}^* - \mathbb{E}[N_{x^n}] \end{bmatrix}. \tag{52}$$

Subtracting Eq. (50) from Eq. (48) we have $\xi = D^{-1}b$, where the LP constraint matrix $D$ is invertible by our assumption that the drifts are nonzero. Finally, letting $\mu^r$ denote the vector of expected rewards, the regret can be expressed as

$$R_T(\texttt{ControlBudget}) = \xi^T \mu^r \tag{53}$$

$$\leq |\xi^T \mu^r| \tag{54}$$

$$\leq \|\xi\|_1 \|\mu^r\|_\infty \tag{55}$$

$$\leq \|D^{-1}\|_1 \|b\|_1 \tag{56}$$

$$\leq C_{\delta_{\text{drift}}} \left( \mathbb{E}[B_T] + \mathbb{E}[N_{x^0}] \right), \tag{57}$$

where $C_{\delta_{\text{drift}}} = O(\delta_{\text{drift}}^{-1})$ is a constant. This completes the proof.

### C.4 Proof of Lemma 3.5

Divide the $T$ rounds into two phases: $P_1 = \{1, \ldots, T - \exp(3/c)\}$ and $P_2 = \{1, \ldots, T\} \setminus P_1$. Note that $P_2$ consists of $\exp(3/c) = O(\exp(\delta_{\text{drift}})) = O(1)$ rounds, where the last equality follows because drifts are bounded by 1. Therefore, the expected number of null arm pulls in this phase is $O(1)$ and it suffices to bound the expected number of null arm pulls in $P_1$.

Consider the following recursive definition for three disjoint sequences of indices $t_i$, $t_i'$ and $t_i''$. Let $t_0 = 0$, and define

$$t_i' = \min\{t > t_i : B_{t-1} \geq \tau_t \text{ or } t - 1 = T\}, \tag{58}$$

$$t_i'' = \min\{t > t_i' : B_{t-1} < \tau_t\}, \tag{59}$$

$$t_{i+1} = \min\{t > t_i'' : B_t < 1\}. \tag{60}$$

We can bound the expected number of rounds in which the budget is less than 1 as

$$\mathbb{E}\left[\sum_{t=1}^{T}\mathbb{1}[B_{t-1} < 1]\right] \tag{61}$$

$$= \sum_{i=0}^{T-1}\mathbf{Pr}\left[t_i \text{ exists }\right]\mathbb{E}\left[\sum_{t=t_i}^{t'_i}\mathbb{1}[B_{t-1} < 1]\right] \tag{62}$$

$$\leq \underbrace{\mathbb{E}\left[\sum_{t=t_0}^{t'_0-1}\mathbb{1}[B_{t-1} < 1]\right]}_{(a)} + \sum_{i=0}^{T-1}\mathbf{Pr}\left[t_{i+1} \text{ exists }|t'_i, t''_i \text{ exist}\right]\underbrace{\mathbb{E}\left[\sum_{t=t_i}^{t'_i-1}\mathbb{1}[B_{t-1} < 1]\right]}_{(a)}. \tag{63}$$

In rounds $\{t_i, \ldots, t'_i-1\}$, the algorithm pulls the null and positive drift arms. The proof of Lemma 3.2 shows that the expected number of null arm pulls in these rounds is at most $\tilde{C}$, where $\tilde{C}$ is defined in Eq. (37). Therefore, we can bound the term (a) in the above inequality by $\tilde{C}$ and we have that

$$\mathbb{E}\left[\sum_{t=1}^{T}\mathbb{1}[B_{t-1} < 1]\right] \leq \tilde{C}\left(1 + \sum_{i=0}^{T-1}\mathbf{Pr}\left[t_{i+1} \text{ exists }|t'_i, t''_i \text{ exist}\right]\right). \tag{64}$$

If $t'_i$ exists, then $B_{t'_i-1} \geq \tau_{t'_i}$. If $t''_i$ exists, then $\tau_{t''_i}-1 \leq B_{t''_i-1} < \tau_{t''_i}$ because (i) $t''_i$ is the first round after $t'_i$ in which the budget is below the threshold; and (ii) the drifts are bounded by 1, so it cannot be lower than $\tau_{t''_i} - 1$. The algorithm pulls the negative drift arm $x^n$ in the rounds $\{t'_i, \ldots, t''_i - 1\}$ and the positive drift arm $x^p$ in the rounds $\{t''_i, \ldots, t_{i+1} - 1\}$. Since the drifts are bounded by 1, it takes at least $\tau_{t''_i} - 2$ rounds for the budget to drop below 1 after repeated pulls of $x^p$. Using this and the observation that the budget dropping below 1 is contained in the event that the total drift in those rounds is nonpositive, we can bound (a) as

$$\mathbf{Pr}\left[t_{i+1} \text{ exists }|t''_i, t'_i \text{ exist}\right] \leq \sum_{q=t''_i+\tau_{t''_i}-2}^{T}\mathbf{Pr}\left[\sum_{t=t''_i+1}^{q}d_t \leq 0\right] \tag{65}$$

$$\leq \sum_{q=t''_i+\tau_{t''_i}-2}^{T}\exp\left(-\frac{1}{2}\delta_{\text{drift}}^2(\tau_{t''_i}-2)\right) \tag{66}$$

$$\leq \sum_{q=t''_i+\tau_{t''_i}-2}^{T}\exp\left(-\frac{1}{2}\delta_{\text{drift}}^2\tau_{t''_i}\right) \tag{67}$$

$$= \sum_{q=t''_i+\tau_{t''_i}-2}^{T}\exp\left(-\frac{1}{2}\delta_{\text{drift}}^2 c\log(T-t''_i)\right) \tag{68}$$

$$\leq \sum_{q=t''_i+\tau_{t''_i}-2}^{T}(T-t''_i)^{-3}, \tag{69}$$

where the second inequality follows from the Azuma-Hoeffding inequality applied to the sequence of drifts sampled from $x^p$ and the last inequality follows because $c \geq \frac{6}{\delta_{\text{drift}}^2}$. The summation is over at most $T - t''_i$ terms because there are at most $T - t''_i$ rounds left after round $t''_i$. Therefore, we have that

$$\mathbf{Pr}\left[t_{i+1} \text{ exists }|t''_i, t'_i \text{ exist}\right] \leq (T-t''_i)^{-2}. \tag{70}$$

Substituting this in Eq. (64), we have that

$$\mathbb{E}\left[\sum_{t=1}^{T}\mathbb{1}[B_{t-1}<1]\right] \leq \tilde{C}\left(1+\sum_{i=0}^{T-1}\mathbf{Pr}\left[t_{i+1}\text{ exists }|t'_i,t''_i\text{ exist}\right]\right) \tag{71}$$

$$\leq \tilde{C}\left(1+\sum_{i=0}^{T-1}(T-t''_i)^{-2}\right) \tag{72}$$

$$\leq \tilde{C}\left(1+\sum_{i=0}^{\infty}(T-t''_i)^{-2}\right) \tag{73}$$

$$\leq \tilde{C}\left(1+\frac{\pi^2}{6}\right). \tag{74}$$

This completes the proof.

### C.5 Proof of Lemma 3.6

Let $E_q$ denote the event that the negative drift arm $x^n$ is pulled consecutively in exactly the last $q$ rounds, i.e., $x_t = x^n$ for all $t \geq T - q + 1$ and $x_t \in \{x^0, x^p\}$ for $t = T - q$ (if $q \neq T$). Note that the events $(E_q : q = 0, \ldots, T)$ are disjoint. Let $S_q$ denote the event that the total drift in the last $q$ pulls of $x^n$ is greater than $\frac{1}{2}\mu^d_{x^n}q$, i.e., $\sum_{t \geq T-q+1} d_t > \frac{1}{2}\mu^d_{x^n}q$. We can upper bound the expected leftover budget by conditioning on these events as follows.

$$\mathbb{E}[B_T] = \sum_{q=0}^{T}\mathbf{Pr}[E_q]\,\mathbb{E}[B_T|E_q] \tag{75}$$

$$\leq \sum_{q=0}^{T}\mathbb{E}[B_T|E_q] \tag{76}$$

$$= \sum_{q=0}^{T}\underbrace{\mathbb{E}[B_T|E_q,S_q]}_{(a)}\,\underbrace{\mathbf{Pr}[S_q|E_q]}_{(b)} + \underbrace{\mathbb{E}[B_T|E_q,S_q^c]}_{(c)}\,\underbrace{\mathbf{Pr}[S_q^c|E_q]}_{(d)}. \tag{77}$$

If $q = 0$, then the expected leftover budget is trivially at most a constant. We can bound the four terms for $q \geq 1$ as follows:

(a) We have
$$\mathbb{E}[B_T|E_q,S_q] \leq c\log q + q \tag{78}$$
because (i) `ControlBudget` (Algorithm 1) pulls $x^0$ or $x^p$ in round $T - q$ if $B_{T-q-1} < \tau_{T-q} = c\log q$; and (ii) conditioned on the event $S_q$, the total drift in the last $q$ rounds can be at most $q$ as the drifts are bounded by 1.

(b) We have
$$\mathbf{Pr}[S_q|E_q] \leq \exp\left(-\frac{1}{16}(\mu^d_{x^n})^2 q\right) \tag{79}$$
because (i) the sequence of drifts observed from $q$ pulls of the negative drift arm $x^n$ is a supermartingale difference sequence; and (ii) by the Azuma-Hoeffding inequality, the probability the sum $S_q$ is greater than half its expected value is at most $\exp\left(-\frac{1}{16}(\mu^d_{x^n})^2 q\right)$.

(c) We have
$$\mathbb{E}[B_T|E_q,S_q^c] \leq \left(c\log q + \frac{1}{2}\mu^d_{x^n}q\right) \tag{80}$$
because (i) `ControlBudget` (Algorithm 1) pulls $x^0$ or $x^p$ in round $T - q$ if $B_{T-q-1} < \tau_{T-q} = c\log q$; and (ii) conditioned on the event $S_q^c$, the total drift in the last $q$ rounds can be at most $\frac{1}{2}\mu^d_{x^n}q$.

(d) We have

$$\mathbf{Pr}[S_q^c | E_q] \leq 1 \tag{81}$$

trivially.

Therefore,

$$\mathbb{E}[B_T] \leq \underbrace{\sum_{q=0}^{T} (c \log q + q) \exp\left(-\frac{1}{16}(\mu_{x^n}^d)^2 q\right)}_{(e)} + \underbrace{\left(c \log q + \frac{1}{2}\mu_{x^n}^d q\right)}_{(f)}. \tag{82}$$

This summation is a constant in terms of $T$:

1. Term (e) is a constant because $c \log q < q$ for $q$ large enough and $\sum_{q=1}^{\infty} q \exp(-aq)$ converges to $\exp(a)(1 - \exp(a))^{-2}$.

2. Term (f) is a constant because this term is negative for $q$ large enough as $\mu_{x^n}^d < 0$ and is maximized at $q = \frac{2c}{|\mu_{x^n}^d|}$.

Finally, we can bound the expected leftover budget as

$$\mathbb{E}[B_T] \leq \tilde{C} = \tilde{O}\left(\left(1 - \exp\left(\frac{\delta_{\text{drift}}^2}{16}\right)\right)^{-2} + \frac{1}{\delta_{\text{drift}}^2}\right), \tag{83}$$

where the last equality follows when $c \geq \frac{6}{\delta_{\text{drift}}^2}$. This completes the proof.

# D  Proofs for Section 3.2

## D.1  Proof of Lemma 3.9

It suffices to show that $\gamma = \frac{\sigma_{\min} \min\{\delta_{\text{support}}, \delta_{\text{slack}}\}}{4m}$ is a feasible solution the Eq. (8).

First, we show that $p = D^{-1}(b + \gamma s_t) \geq 0$. For each $x \in X$,

$$\begin{aligned}
e_x^T D^{-1}(b + \gamma s_t) &= e_x^T D^{-1} b + \gamma e_x^T D^{-1} s_t \\
&= p_x^* + \gamma e_x^T D^{-1} s_t \\
&\geq \delta_{\text{support}} - \gamma \|D^{-1} s_t\|_2 \\
&\geq \delta_{\text{support}} - \gamma \frac{1}{\sigma_{\min}} \sqrt{m} \\
&\geq 0.
\end{aligned}$$

Second, we show that for any non-binding resource $j$, $d_j^T D^{-1}(b + \gamma s_t) \geq \frac{\delta_{\text{slack}}}{2}$:

$$\begin{aligned}
d_j^T D^{-1}(b + \gamma s_t) &= \sum_{x \in X} d_j(x, \mu) p_x^* + \gamma d_j^T D^{-1} s_t \\
&\geq \delta_{\text{slack}} - \gamma |d_j^T D^{-1} s_t| \\
&\geq \delta_{\text{slack}} - \gamma \|d_j^T\|_2 \|D^{-1}\|_2 \|s_t\|_2 \\
&\geq \delta_{\text{slack}} - \gamma \frac{1}{\sigma_{\min}} m \\
&\geq \frac{\delta_{\text{slack}}}{2} \\
&\geq \frac{\gamma}{2},
\end{aligned}$$

where the last inequality follows because $\sigma_{\min}, \delta_{\text{slack}}, \delta_{\text{support}} < 1$.

## D.2 Proof of Lemma 3.10

Divide the $T$ rounds into two phases: $P_1 = \{1, \ldots, T - \exp(3/c)\}$ and $P_2 = \{1, \ldots, T\} \setminus P_1$. Note that $P_2$ consists of $\exp(3/c) = O(\exp(\gamma^*)) = O(1)$ rounds, where the last equality follows because $\gamma^*$ is bounded by 1. Therefore, the expected number of null arm pulls in this phase is $O(1)$ and it suffices to bound the expected number of null arm pulls in $P_1$.

We can write the expected number of rounds in which there exists a resource whose budget is less than 1 as

$$\mathbb{E}\left[\sum_{t=1}^{T}\sum_{j\in\mathcal{J}}\mathbb{1}[B_{t-1,j}<1]\right] = \sum_{j\in\mathcal{J}}\mathbb{E}\underbrace{\left[\sum_{t=1}^{T}\mathbb{1}[B_{t-1,j}<1]\right]}_{(a)}. \tag{84}$$

We can bound term (a) above the same way as in the proof of Lemma 3.5 (Appendix C.4) with $\delta_{\text{drift}}$ replaced by $\gamma^*$.. Therefore,

$$\mathbb{E}\left[\sum_{t=1}^{T}\sum_{j\in\mathcal{J}}\mathbb{1}[B_{t-1,j}<1]\right] \leq m\tilde{C}\left(1+\frac{\pi^2}{6}\right), \tag{85}$$

where $\tilde{C}$ is defined in Eq. (37).

## D.3 Proof of Lemma 3.11

Consider an arbitrary resource $j \in J^*$. Recall the vector $s_t$ defined in `ControlBudget` (Algorithm 2). If $i$ denote the row corresponding to resource $j$, then the $i$th entry of $s_t$, denoted by $s_t(i)$, is $-1$ if $B_{t-1,j} < \tau_t$ and $+1$ otherwise.

Let $E_q$ denote the event that the $s_t(i)$ is equal to $-1$ consecutively in exactly the last $q$ rounds, i.e., $s_t(i) = -1$ for all $t \geq T - q + 1$ and $s_t(i) = +1$ for $t = T - q$ (if $q \neq T$). Note that the events $(E_q : q = 0, \ldots, T)$ are disjoint. Let $S_q$ denote the event that the total drift for $j$ in the last $q$ rounds is greater than $\frac{1}{2}(-\gamma^*)q$, i.e., $\sum_{t \geq T-q+1} d_t > \frac{1}{2}(-\gamma^*)q$. We can upper bound the expected leftover budget of resource $j$ by conditioning on these events as follows.

$$\mathbb{E}[B_{T,j}] = \sum_{q=0}^{T}\mathbf{Pr}[E_q]\,\mathbb{E}[B_{T,j}|E_q] \tag{86}$$

$$\leq \sum_{q=0}^{T}\mathbb{E}[B_{T,j}|E_q] \tag{87}$$

$$= \sum_{q=0}^{T}\underbrace{\mathbb{E}[B_{T,j}|E_q, S_q]}_{(a)}\underbrace{\mathbf{Pr}[S_q|E_q]}_{(b)} + \underbrace{\mathbb{E}[B_{T,j}|E_q, S_q^c]}_{(c)}\underbrace{\mathbf{Pr}[S_q^c|E_q]}_{(d)}. \tag{88}$$

If $q = 0$, then the expected leftover budget is trivially at most a constant. We can bound the four terms for $q \geq 1$ as follows:

(a) We have
$$\mathbb{E}[B_{T,j}|E_q, S_q] \leq c\log q + q \tag{89}$$
because (i) `ControlBudget` (Algorithm 2) sets $s_t(i) = +1$ in round $T - q$ if $B_{T-q-1,j} < \tau_{T-q} = c\log q$; and (ii) conditioned on the event $S_q$, the total drift in the last $q$ rounds can be at most $q$ as the drifts are bounded by 1.

(b) We have
$$\mathbf{Pr}[S_q|E_q] \leq \exp\left(-\frac{1}{16}(\gamma^*)^2 q\right) \tag{90}$$
because (i) the sequence of drifts observed in rounds $t \geq T - q + 1$ is a supermartingale difference sequence with $\mathbb{E}[d_{s,j}|d_{T-q+1,j}, \ldots, d_{s-1,j}] \leq -\gamma^*$; and (ii) by the Azuma-Hoeffding inequality, the probability the sum $S_q$ is greater than half its expected value is at most $\exp\left(-\frac{1}{16}(\gamma^*)^2 q\right)$.

(c) We have

$$\mathbb{E}[B_{T,j}|E_q, S_q^c] \leq \left(c\log q + \frac{1}{2}(-\gamma^*)q\right) \tag{91}$$

because (i) ControlBudget (Algorithm 2) sets $s_t(i) = +1$ in round $T - q$ if $B_{T-q-1,j} < \tau_{T-q} = c\log q$; and (ii) conditioned on the event $S_q^c$, the total drift in the last $q$ rounds can be at most $\frac{1}{2}(-\gamma^*)q$.

(d) We have

$$\mathbf{Pr}[S_q^c|E_q] \leq 1 \tag{92}$$

trivially.

Therefore,

$$\mathbb{E}[B_{T,j}] \leq \underbrace{\sum_{q=0}^{T} (c\log q + q)\exp\left(-\frac{1}{16}(\gamma^*)^2 q\right)}_{(e)} + \underbrace{\left(c\log q + \frac{1}{2}(-\gamma^*)q\right)}_{(f)}. \tag{93}$$

This summation is a constant in terms of $T$:

1. Term (e) is a constant because $c\log q < q$ for $q$ large enough and $\sum_{q=1}^{\infty} q\exp(-aq)$ converges to $\exp(a)(1 - \exp(a))^{-2}$.

2. Term (f) is a constant because this term is negative for $q$ large enough and is maximized at $q = \frac{2c}{\gamma^*}$.

Finally, we can bound the expected leftover budget as

$$\mathbb{E}[B_{T,j}] \leq \tilde{C} = \tilde{O}\left(\left(1 - \exp\left(\frac{\gamma^{*2}}{16}\right)\right)^{-2} + \frac{1}{\gamma^{*2}}\right), \tag{94}$$

where the last equality follows when $c \geq \frac{6}{\gamma^{*2}}$. This completes the proof.

# E    Proofs for Section 4

## E.1    Proof of Lemma 4.1

It suffices to show that the complement of the clean event occurs with probability at most $5mT^{-2}$.

For (i) in the definition of the clean event (Definition 4.2), by taking a union bound over the components of the outcome vector and using Azuma-Hoeffding inequality, we have

$$\mu_x^o \notin [\mathsf{LCB}_t(x), \mathsf{UCB}_t(x)] \leq 2(m+1)\exp\left(-2n_t(x)\mathsf{rad}_t(x)^2\right) \tag{95}$$

$$\leq 4m\exp\left(-2n_t(x)\frac{8\log T}{n_t(x)}\right) \tag{96}$$

$$\leq 4mT^{-2}. \tag{97}$$

For (ii) in the definition of the clean event (Definition 4.2), a similar approach works. Let $S_{n,j}$ denote the sum of the drifts for resource $j \in \mathcal{J}$ after $n$ pulls of the null arm $x^0$. By the union bound and

Azuma-Hoeffding inequality,

$$\mathbf{Pr}\left[\exists j \in \mathcal{J} \text{ s.t. } S_{n,j} < w\right] \leq m \exp\left(-\frac{1}{4}w\mu_{x^0}^d\right) \tag{98}$$

$$\leq m \exp\left(-\frac{1}{4}w\delta_{\text{drift}}\right) \tag{99}$$

$$\leq m \exp\left(-\frac{1}{4}\frac{1024km^2\log T}{\delta_{\text{drift}}^2\sigma_{\min}^2}\delta_{\text{drift}}\right) \tag{100}$$

$$= m \exp\left(-\frac{256km^2\log T}{\delta_{\text{drift}}\sigma_{\min}^2}\right) \tag{101}$$

$$\leq m \exp\left(-256km^2\log T\right) \tag{102}$$

$$\leq m \exp\left(-256\log T\right) \tag{103}$$

$$\leq mT^{-2}, \tag{104}$$

where Eq. (102) follows because $\delta_{\text{drift}} \in (0,1]$ and $\sigma_{\min} \in (0,1)$. This shows that the probability of the complement of the clean event is at most $5mT^{-2}$ and completes the proof.

## E.2 Proof of Lemma 4.2

We will prove the lemma for $\mathsf{OPT_{LP}}$ because the other cases are similar. Simplifying and overloading notation for this proof, we denote the probability simplex over $k$ dimensions as $\Delta_k$, and the vector of expected rewards, the matrix of expected drifts and the right-hand side of the budget constraints as

$$r = \begin{bmatrix} \mu_1^r \\ \vdots \\ \mu_k^r \end{bmatrix}, \ D = \begin{bmatrix} \mu_1^{d,1} & \cdots & \mu_k^{d,1} \\ & \ddots & \\ \mu_1^{d,m} & \cdots & \mu_k^{d,m} \end{bmatrix}, \ b = -\frac{B}{T}\mathbf{1}. \tag{105}$$

We will use $\bar{r}$ and $\bar{D}$ to denote the empirical versions of the rewards and drifts. We can write

$$\mathsf{OPT_{LP}} = \max_{p \in \Delta_k} r^T p \qquad\qquad \text{s.t. } Dp \geq b,$$

$$\mathsf{UCB}_t(\mathsf{OPT_{LP}}) = \max_{q \in \Delta_k} (\bar{r} + \mathsf{rad}_t)^T q \qquad\qquad \text{s.t. } (\bar{D} + \mathsf{rad}_t)q \geq b$$

$$\leq \max_{q \in \Delta_k} (r + 2\mathsf{rad}_t)^T q \qquad\qquad \text{s.t. } (D + 2\mathsf{rad}_t)q \geq b$$

$$\leq 2\mathsf{rad}_t + \max_{q \in \Delta_k} r^T q \qquad\qquad \text{s.t. } Dq \geq b - 2\mathsf{rad}_t,$$

where the second-last inequality follows because we are conditioning on the clean event. Therefore, using $D'$ and $b'$ to denote the submatrix and subvector corresponding to the binding constraints, we have

$$\mathsf{UCB}_t(\mathsf{OPT_{LP}}) - \mathsf{OPT_{LP}} \leq 2\mathsf{rad}_t + |r^T p - r^T q|$$

$$\leq 2\mathsf{rad}_t + |r^T(D')^{-1}b' - r^T(D')^{-1}(b' - 2\mathsf{rad}_t)|$$

$$\leq 2\mathsf{rad}_t + \|r\|_2\|(D')^{-1}\|_2\|2\mathsf{rad}_t\|_2$$

$$\leq 2\mathsf{rad}_t + 2m\mathsf{rad}_t\frac{1}{\sigma_{\min}}$$

$$\leq \frac{4m}{\sigma_{\min}}\mathsf{rad}_t,$$

where last inequality follows because $\sigma_{\min} < 1 \leq m$. Since the LCB is defined by subtracting $\mathsf{rad}_t$ from the empirical means, we obtain the same upper bound on $\mathsf{OPT_{LP}} - \mathsf{LCB}_t(\mathsf{OPT_{LP}})$ and using the triangle inequailty completes the proof.

## E.3 Proof of Theorem 4.1

Since the complement of the clean event occurs with probability at most $O(mT^{-2})$ and contributes $O(T)$ to the regret, it suffices to bound the regret conditioned on the clean event. So, condition on the

clean event for the rest of the proof. Phase one contributes at most

$$O\left(\frac{km^2}{\min\{\delta_{\mathrm{drift}}^2, \sigma_{\min}^2\}\Delta^2}\right) \cdot \log T \tag{106}$$

to the regret by Corollary 4.2. Phase two contributes at most

$$O\left(\frac{k}{\gamma^{*2}}\right) \log T \tag{107}$$

to the regret.

Observe that after phase two, $\mathsf{rad}_t(x) \leq \frac{\gamma^{*2}}{2}$ for all $x \in X^*$. Combining this with Eqs. (8) and (15) to (17), we have that $(\gamma^*, D^{-1}(b + \gamma^* s_t))$ is a feasible solution to the optimization problem solved by `ExploreThenControlBudget` (Algorithm 3). Therefore, $(\gamma_t, p_t)$ ensure that there is drift of magnitude at least $\frac{\gamma^*}{8}$ in the "correct directions". As noted in the end of Section 3.2, the regret analysis of `ControlBudget` (Algorithm 2) requires the algorithm to know $X^*$, $J^*$, and find a probability vector $p_t$ that ensures drifts bounded away from zero in the "correct directions". Therefore, by Theorem 3.2, phase three contributes at most $\tilde{C}'$ to the regret, where $\tilde{C}$ is the constant in Theorem 3.2. Combining the contribution from the three phases, we have that

$$R_T(\texttt{ExploreThenControlBudget}) \leq \tilde{C} \cdot \log T, \tag{108}$$

where $\gamma^*$ (defined in Lemma 3.9) and $\tilde{C}$ are constants with

$$\tilde{C} = O\left(\frac{km^2}{\min\{\delta_{\mathrm{drift}}^2, \sigma_{\min}^2\}\Delta^2} + k(\gamma^*)^{-2} + \tilde{C}'\right). \tag{109}$$

### E.4 Proof of Theorem 4.2

Note that BwK is not automatically a special case of our model because of our assumption that the null arm has strictly positive drift for every resource. In this section we present a reduction from BwK with $\frac{B}{T}$ bounded away from 0 to our model. We show that our results imply a logarithmic regret bound for BwK under certain assumptions.

**Reduction**   Assume we are given an instance of BwK with $\frac{B}{T} \geq \delta_{\mathrm{drift}} > 0$. (Existing results on logarithmic regret for BwK also assume the ratio of the initial budget to the time horizon is bounded away from 0 [14].) We will reduce the given BwK instance to a problem in our model. The reduction initializes an instance of `ExploreThenControlBudget` (Algorithm 3) running in a simulated environment with the same set of arms as in the given BwK instance, plus an additional null arm whose drift is equal to $\delta_{\mathrm{drift}}$ deterministically for each resource. The reduction will maintain two time counters: $t_a$ is the actual number of time steps that have elapsed in the BwK prlblem, and $t_s$ is the number of time steps that have elapsed in the simulation environment in which Algorithm 3 is running. Likewise, there are two vectors that track the remaining budget: $B_a$ is the remaining budget in the actual BwK problem our reduction is solving, while $B_s$ is the remaining budget in the simulation environment. These two budget vectors will always be related by the equation

$$B_s = B_a - T\delta_{\mathrm{drift}}\mathbf{1} + t_s\delta_{\mathrm{drift}}\mathbf{1}. \tag{110}$$

In particular, the initial budget of each resource is initialized (at simulated time $t_s = 0$) to $B - T\delta_{\mathrm{drift}}$.

Each step of the reduction works as follows. We call Algorithm 3 to simulate one time step in the simulated environment. If Algorithm 3 recommends to pull a non-null arm $x$, we pull arm $x$, increment both of the time counters ($t_a$ and $t_s$), and update the vector of remaining resource amounts, $B_a$, according to the resources consumed by arm $x$. If Algorithm 3 recommends to pull the null arm, we do not pull any arm, and we leave $t_a$ and $B_a$ unchanged; however, we still increment the simulated time counter $t_s$. Finally, regardless of whether a null or non-null arm was pulled, we update $B_s$ to satisfy Eq. (110).

**Correctness**   Since the reduction pulls the same sequence of non-null arms as Algorithm 3 until the BwK stopping condition is met and the additional pulls of the null arm in the simulation environment yield zero reward, the total reward in the actual BwK problem equals the total reward earned in the

simulation environment at the time when the BwK stopping condition is met and the reduction ceases running. Since Algorithm 3 maintains the invariant that $B_s$ is a nonnegative vector, Eq. (110) ensures that $B_a$ will also remain nonnegative as long as $t_s \geq T$ must hold. Theorem 4.1 ensures that the total expected reward earned in the simulation environment and hence, also in the BwK problem itself, is bounded below by $T \cdot \mathsf{OPT}_{\mathsf{LP}} - \tilde{C} \cdot \log T$, where $\tilde{C}$ is the constant in Theorem 4.1 and $\mathsf{OPT}_{\mathsf{LP}}$ denotes the optimal value of the LP relaxation (Eq. (1)) for the simulation environment.

We would like to show that this implies the regret of the reduction (with respect to the LP relaxation of BwK) is bounded by $\tilde{C} \cdot \log T$. To do so, we must show that the LP relaxations of the original BwK problem and the simulation environment have the same optimal value. Let $\mu_x^r$ and $\mu_x^{d,j}$ denote the expected reward and expected drifts in the actual BwK problem with arm set $\mathcal{X}$, and let $\hat{\mu}_x^r$ and $\hat{\mu}_x^{d,j}$ denote the expected reward and drifts in the simulation environment with arm set $\mathcal{X}^+ = \mathcal{X} \cup \{x^0\}$. The two LP formulations are as follows.

$$
\begin{array}{ll}
\max\limits_{p} & \sum\limits_{x \in \mathcal{X}} p_x \mu_x^r \\
\text{s.t.} & \sum\limits_{x \in \mathcal{X}} p_x \mu_x^{d,j} \geq -\frac{B}{T} \quad \forall j \in \mathcal{J}, \\
& \sum\limits_{x \in \mathcal{X}} p_x \leq 1 \\
& p_x \geq 0 \qquad \forall x \in \mathcal{X}.
\end{array}
\qquad
\begin{array}{ll}
\max\limits_{p} & \sum\limits_{x \in \mathcal{X}} p_x \hat{\mu}_x^r \\
\text{s.t.} & \sum\limits_{x \in \mathcal{X}} p_x \hat{\mu}_x^{d,j} \geq -\frac{B}{T} - \delta_{\text{drift}} \quad \forall j \in \mathcal{J}, \\
& \sum\limits_{x \in \mathcal{X}^+} p_x = 1 \\
& p_x \geq 0 \qquad \forall x \in \mathcal{X}^+.
\end{array}
$$

The differences between the two LP formulations lie in substituting $\hat{\mu}$ for $\mu$, substituting $\mathcal{X}^+$ for $\mathcal{X}$, and transforming the inequality constraint $\sum_{x \in \mathcal{X}} p_x \leq 1$ into an equality constraint $\sum_{x \in \mathcal{X}^+} p_x = 1$. We know that $\mu_x^r = \hat{\mu}_x^r$ for every $x \in \mathcal{X}$ and $\hat{\mu}_{x^0}^r = 0$. Furthermore, $\hat{\mu}_x^{d,j}$ denotes the expected drift of resource $j$ in the simulation environment when arm $x$ is pulled. This can be written as the sum of two terms: drift $\mu_x^{d,j}$ is the expectation of the (non-positive) quantity added to the $j$th component of budget vector $B_a$ when pulling arm $x$ in the actual BwK environment; in addition to this non-positive drift, there is a deterministic positive drift of $\delta_{\text{drift}}$ due to incrementing the simulation time counter $t_s$ and recomputing $B_s$ using Eq. (110). Hence, $\hat{\mu}_x^{d,j} = \mu_x^{d,j} + \delta_{\text{drift}}$ for all $x \in \mathcal{X}$ and $j \in \mathcal{J}$. Furthermore, $\hat{\mu}_{x^0}^{d,j} = \delta_{\text{drift}}$. Hence, for any vector $\vec{p}$ representing a probability distribution on $\mathcal{X}^+$, we have

$$
\sum_{x \in \mathcal{X}^+} p_x \hat{\mu}_x^{d,j} = \left( \sum_{x \in \mathcal{X}} p_x \hat{\mu}_x^{d,j} \right) + \delta_{\text{drift}}. \tag{111}
$$

Accordingly, a vector $\vec{p}$ satisfies the constraints of the BwK LP relaxation above if and only if the probability vector on $\mathcal{X}^+$ obtained from $\vec{p}$ by setting $p_{x^0} = 1 - \sum_{x \in \mathcal{X}} p_x$ satisfies the constraints of the second LP relaxation above. This defines a one-to-one correspondence between the sets of vectors feasible for the two LP formulations. Furthermore, this one-to-one correspondence preserves the value of the objective function because $\hat{\mu}_x^r = \mu_x^r$ for $x \in \mathcal{X}$ and $\hat{\mu}_{x^0}^r = 0$. Thus, the optimal value of the two linear programs is the same. This completes the proof.

## F  Experiments

In this section we present some simple experimental results. [5] For simplicity, we only consider Bernoulli distributions, i.e., rewards are supported on $\{0, 1\}$, a positive drift arm's drifts are supported on $\{0, 1\}$, and a negative drift arm's drifts are supported on $\{0, -1\}$. We generate the data for the experiments as follows:

- Fig. 1 plot (a): We set $T = 25k, B = 0, n = 2$ and $m = 1$. The expected reward and drifts for the arms are: $(0; 0.1), (0.8; 0.4)$. The LP solution is supported on a single positive drift arm.

- Fig. 1 plot (b): We set $T = 25k, B = 400, n = 2$ and $m = 1$. The expected reward and drifts for the arms are: $(0; 0.4), (0.8; -0.3)$. The LP solution is supported on the null arm and the negative drift arm.

- Fig. 1 plot (c): We set $T = 25k, B = 400, n = 3$ and $m = 1$. The expected reward and drifts for the arms are: $(0; 0.4), (0.8; -0.3), (0.1; 0.3)$. The LP solution is supported on the positive drift arm and the negative drift arm.

---

[5]The code and data are available at `https://github.com/raunakkmr/non-monotonic-resource-utilization-in-the-bandits-with-knapsacks-problem-code`.

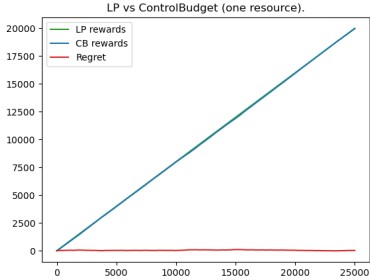

(a) `ControlBudget` with one resource - case 1 (positive drift arm)

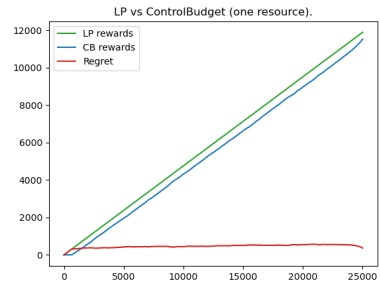

(b) `ControlBudget` with one resource - case 2 (null plus negative drift arm)

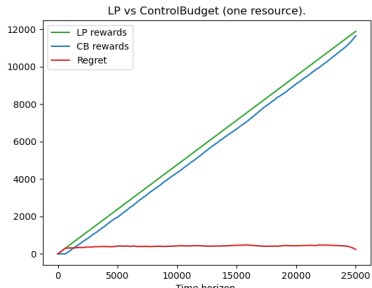

(c) `ControlBudget` with one resource - case 3 (positive plus negative drift arm)

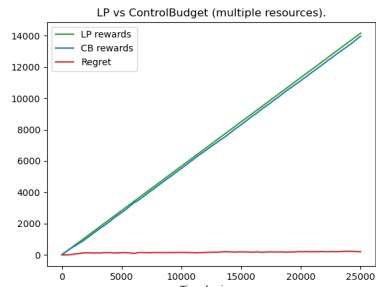

(d) `ControlBudget` with multiple resources

Figure 1: Regret of `ControlBudget` on a variety of test cases.

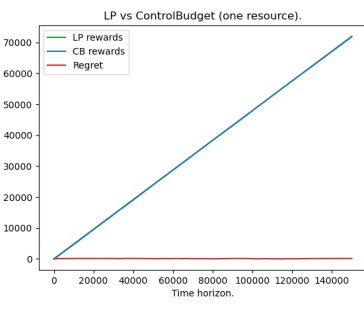

(a) `ControlBudget`

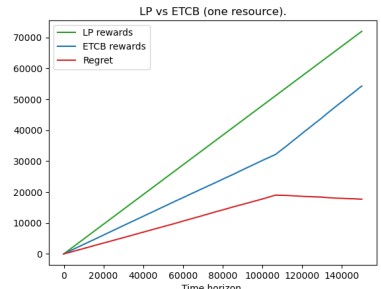

(b) `ExploreThenControlBudget`

Figure 2: Regret of `ControlBudget` and `ExploreThenControlBudget` on the same test case. We modify `ExploreThenControlBudget` to use the empirical means instead of UCB/LCB estimates for phase one as described in Appendix F.

- Fig. 1 plot (d): We set $T = 25k, B = 3, n = 3$ and $m = 2$. The expected reward and drifts for the arms are: $(0; 0.1, 0.08), (0.8; -0.2, -0.25), (0.1; 0.4, 0.5)$. The LP solution is supported on the positive drift arm and the negative drift arm.

- Fig. 2 plot (a) and (b): We set $T = 150k, B = 10, n = 3$ and $m = 1$. The expected reward and drifts for the arms are: $(0; 0.9), (0.8; -0.6), (0.1; 0.7)$. The LP solution is supported on the positive drift arm and the negative drift arm.

As our plots show (Fig. 1), our MDP policy, `ControlBudget`, performs quite well and achieves constant regret.

Our learning algorithm does not perform as well empirically due to large constant factors. Specifically, the number of rounds required for the confidence radius to be small enough for phase one to

successfully identify $X^*$ and $J^*$ is too large. In our simple test cases, if we simply consider the empirical means, which are very close to the true means, instead of the UCB/LCB estimates for phase one, then the learning algorithm performs as expected: it achieves logarithmic regret by spending a logarithmic number of rounds identifying $X^*$ and $J^*$, and achieves constant regret thereafter (Fig. 2).