# OpenReview forum: "Non-monotonic Resource Utilization in the Bandits with Knapsacks Problem"
_NeurIPS.cc/2022/Conference — NeurIPS 2022 Accept_

### Official Review · Reviewer_S3ZC · 2022-07-13

**Rating:** 7
**Confidence:** 3
**Soundness:** 4 excellent
**Presentation:** 3 good
**Contribution:** 3 good

**Summary:**

Authors introduce a generalization of the Bandits with Knapsack problem in which resources can be stochastically replenished over time. They introduce an algorithm which achieves constant regret with respect to an upper bound of the optimal reward both for the single-resource case and the multi-resource case, which involves non-trivial analysis. They then introduce a learning algorithm that they prove has log(T) regret. They finish by showing that BwK can be reduced to their generalization and as such, the same bounds can apply, matching the state of the art.

**Questions:**


- Can the authors please provide more motivation for why this generalization of BwK is significant?
- Can the authors please provide either minimal empirical results or a commentary on the practical usefulness of their theoretical results, especially as related to the assumptions made in section 2.3?
- Can the authors comment on how their algorithm and results adapt to the case of deterministic costs, e.g., Case 2 of Flajolet and Jaillet, which also has many real-world applications?



**Limitations:**

The authors acknowledge that their work does not achieve the optimal regret bound. They do not discuss the social implications of their work, which is ok for this mostly theoretical submission.

**Strengths And Weaknesses:**


Strengths:
 - Theoretical analysis is thorough, including some relatively non-intuitive results, e.g., Lemma 3.5 which shows that the proper choice of budget threshold can eliminate the regret's dependency on the horizon T. Main results are:
   - Constant regret achieved in the case where reward and cost distributions are known. They achieve mainly by making assumptions necessary to ensure that the optimal solution can reduce to cases amenable to analysis, i.e., there is a subset of arms+resources manipulated in the upper bound solution, and that each arm is "positive" or "negative" drift with respect to a given resource -- these allow analysis of small number of individual cases where the challenge is to compute in expectation how often the authors' proposed policy plays arms that deviate from the upper bound solution.
   - Log(T) regret achieved in the online learning case, via building off of pieces of a previously used multi-stage learning approach which identifies and improves confidence about the "best arms" in the first stages in log(T) time, then runs the optimal algorithm in the final stage.
   - The reduction from BwK is helpful in grounding their results in current literature
 - The paper is relatively clearly written for what is a very math-dense submission. Authors do a nice job providing intuitive explanations within their proof sketches of how they arrived at certain key results, e.g., lines 183-190.
 - This seems to be a novel bandits problem.

Weaknesses:
 - In my view, the authors do not do enough to motivate the problem they introduce. The example they give is "in a dynamic pricing application a seller may receive shipments that increase their inventory level," but it is not clear that the author's model is the proper one even for this example. I.e., is it reasonable to assume that shipments would arrive stochastically and without respect to current supply? More examples should be given, or citations suggesting that this generalization will have impact on the theory community would be helpful. Without a stronger motivation for the problem, it is challenging to evaluate the significance/potential impact of the work.
 - In my view, it is a weakness not to provide any empirical results. These would help readers understand the current practical usefulness of the theoretical results.
 - Authors also do not provide much commentary on how restrictive their assumptions in section 2.3 are, which also limits understanding the practical usefulness.

---

> ### Author Response · Authors · 2022-08-02
> **Response**
>
> Thank you for your feedback. Here are the responses to your questions.
> ## Response to question 1
> [Related to response to question 1 from reviewer 5cwS]
>
> Our paper constitutes a first attempt to modify the BwK problem to incorporate non-monotonic resource utilization. We believe this generalization is significant because the phenomenon of resource replenishment is quite prevalent in learning settings. In the paper we gave the example of a seller using a bandit algorithm for dynamic pricing, in an environment where the inventory level decreases when items are sold and increases when shipments are received. Another example is a scheduling problem with multiple types of servers (arms) and one or more types of jobs (resources) that arrive stochastically and remain in the system until they are completed. For example, one may think of the servers as doctors with different specialties, and the jobs as patients seeking treatment until their problem is resolved. When a server completes a job successfully, a reward is earned and the supply of that job type decreases by 1. At the same time, new jobs of each type may enter the system according to a stochastic arrival process.
>
> You also ask, "Is it reasonable to assume that shipments would arrive stochastically and without respect to current supply?" This is not quite what we assume. We assume that the distribution of the drift (i.e., replenishment minus consumption) **conditional on the arm pulled** is conditionally independent of the past history (and hence, the current supply). In fact, our proofs remain valid even if the distribution of the drift (i.e., replenishment minus consumption) can depend on past history (and hence, on the current supply) provided that the conditional expectation of the drift is nonzero and independent of the past history. (Since our proofs rely on the Azuma-Hoeffding inequality, we need the above assumption on the conditional expectation to hold.)
>
> If one simply wants to assume that a seller orders shipments to restore inventory levels when the current supply is low and chooses not to order shipments when the supply is high, one can incorporate this into the model by including an arm (or set of arms) representing the act of ordering shipments. In the learning setting, this requires the learner to know which arms correspond to the act of ordering shipments and we believe this is a reasonable assumption.
>
> ## Response to question 2
> [Similar response to question 3 from reviewer RYqt]
> Some experimental results are now included in the rebuttal revision in Appendix E.
>
> We acknowledge that our learning algorithm is not very practical because of the constant factors, although using the empirical means instead of the true means seems to work on simple test cases. However, the main contribution of our work is designing a novel MDP policy, ControlBudget, and showing that it has constant regret. This also lets any future learning algorithms to be compared directly to the LP instead of the optimal policy because, as our theorems in section 3 show, the gap between the optimal policy and LP is a constant.
>
> ## Response to question 3
> If both rewards and drifts are deterministic, then this is a path planning problem that can be solved using the Bellman-Ford algorithm.
>
> If the rewards are stochastic but drifts are deterministic, the algorithm becomes simpler but the regret bound is essentially the same with minor changes as described below.
>
> Note that once we identify the arms and binding resources that constitute the unique optimal solution, $X^*$ and $J^*$, then we can play the MDP policy, ControlBudget, instead of phase 3 of the learning algorithm, ExploreThenControlBudget (section 4, algorithm 3). This is because ControlBudget only relies on the expected drifts, which in this case are known once we play each once and observe the deterministic drifts. This achieves constant regret. It remains to analyze the number of rounds required to identify $X^*$ and $J^*$. From the proof of lemma 4.2 (section D.2 in the Appendix) and the fact that the drifts are deterministic, the difference between the UCB and the true value of the LP is at most twice the confidence radius. Therefore, we need to play each arm to ensure that $ 8 rad_t < \Delta $. Therefore, we can bound the length of phase one by $O(\frac{k m^2}{\delta_{\text{drift}}^2 \Delta^2} \log T)$ (eliminating the dependence on $\sigma_{\min}$). We can also eliminate phase two.

---

### Official Review · Reviewer_RYqt · 2022-07-14

**Rating:** 6
**Confidence:** 4
**Soundness:** 3 good
**Presentation:** 3 good
**Contribution:** 3 good

**Summary:**

This paper introduced a generalization of the stochastic bandit with knapsacks (BwK), where in each round, the decision maker observes an outcome consisting of a reward and a vector of resource drifts that can be positive, or negative, or zero. The authors considered two problems depending on whether the knowledge of true outcome distributions is available. If the knowledge is available, the authors developed a Markov decision process (MDP) policy that has constant regret against a linear program (LP) relaxation. Based on this result, the authors developed a learning algorithm that has logarithmic regret against the same LP relaxation when the distribution knowledge is not available. The authors also showed that their regret bound matches a reduction from BwK to their model.

**Questions:**

- The key difference in this paper is that there exists a null arm $x_0$ such that, if played, the expected drift for each resource is positive. This seems to suggest that the resource will increase linearly on average due to the expected positive drift when the null arm is played. To some extent, the resource replenishment rate is quite fast and hence this modeling assumption is quite strong. With this modeling assumption, one can use this null arm to quickly increase the resource budget to sustain the bandit learning. As a result, the strong constant and logarithmic regret under this modeling assumption is not very surprising. Can this assumption be weakened?

- In the ExploreThenControlBudget learning algorithm, the first while-loop is quite similar to the literature and the associated regret analysis appears to be standard and not surprising. It's unclear what is the major difference, challenges, novelty, and new contributions in this part. It would be great if the authors could clarify the challenges and their contributions in this part.

- Although I understand this paper is of theoretical nature, it's still worth providing some numerical experiments to verify the performance of the proposed learning algorithms, particularly given the fact some of the constants in the theoretical results are likely not optimal (e.g., 4096 in Definition 4 and 2048 in Corollary 4.1). Could the authors provide some experimental results?

**Limitations:**

This paper is theoretical and not relevant to negative societal imapcts.

**Strengths And Weaknesses:**

Strengths:

+ The authors proposed a generalized BwK model with non-monotonic resource utilization that has several interesting applications.
+ This paper proposed two policies that achieve constant and logarithmic regrets for cases with or without true outcome distributions, respectively.

Weaknesses:

- The rationale of the null arm modeling assumption appears to be somewhat strong and unclear.
- The UCB/LCB-type learning and regret analysis techniques appear to be standnard.
- Lack of experiments to verify the proposed algorithms.

---

> ### Author Response · Authors · 2022-08-02
> **Response**
>
> Thank you for your feedback. Here are the responses to your questions.
> ## Response to question 1
> In any model in which resources can be consumed and/or replenished over time, one must specify what happens when the budget of one (or more) resources reaches zero. The original bandits with knapsacks problem assumes than when this happens, the process of learning and gaining rewards ceases. The key distinction between that model and ours is that we instead assume the learner is allowed remain idle until the supply of every resource becomes positive again, at which point the learning process recommences. The null arm in our paper is intended to represent this option to remain idle and wait for resource replenishment. In order for these idle periods to have finite length almost surely, a minimal assumption is that when the null arm is pulled, for each resource there is a positive probability that the supply of the resource increases. We make the stronger assumption that for each resource, the expected change in supply is positive when the null arm is pulled. In fact, our results for the MDP setting hold under the following more general assumption: there exists a probability distribution over arms, such that when a random arm is sampled from this distribution and pulled, the expected change in the supply of each resource is positive. In the following, we refer to this as Assumption PD (for "positive drift").
>
> To see that our results for the MDP setting continue to hold under Assumption PD (i.e., even if one doesn't assume that the null arm itself is guaranteed to yield positive expected drift for each resource) simply modify Algorithms 1 and 2 so that whenever they pull the null arm in a time step when the supply of each resource is at least 1, the modified algorithms instead pull a random arm sampled from the probability distribution over arms that guarantees positive expected drift for every resource. As long as the constant $\delta_{\mathrm{drift}}$ is less than or equal to this positive expected drift, the modification to the algorithms does not change their analysis. We believe it's likely that our learning algorithm (Algorithm 3) could similarly be adapted to work under Assumption PD, but it would be less straightforward because the positive-drift distribution over arms would need to be learned.
>
> When Assumption PD is violated, the problem becomes much more similar to the Bandits with Knapsacks problem. To see why, consider a two-player zero-sum game in which the row player chooses an arm $x$, the column player chooses a resource $j$, and the payoff is the expected drift of that resource when that arm is pulled, $\mu_x^{d,j}$. Assumption PD is equivalent to the assertion that the value of the game is positive; the negation of Assumption PD means that the value of the game is negative. By the Minimax Theorem, this means there is a convex combination of resources (i.e., a mixed strategy for the column player) such that the weighted-average supply of these resources is guaranteed to experience non-positive expected drift, no matter which arm is pulled. Either the expected drift is zero --- we prove in Appendix A of the supplementary material that regret $O(\sqrt{T})$ is unavoidable in this case --- or the expected drift is strictly negative, in which case the weighted-average resource supply inevitably dwindles to zero no matter which arms the learner pulls. In either case, the behavior of the model is qualitatively different when Assumption PD does not hold.
>
> ## Response to question 2
> The main contribution of our work is the MDP policy (section 3, algorithms 1 and 2). We present a novel algorithm and prove that it has constant regret with respect to the LP. Such a result was not known even for BwK. Based on this, it is fairly straightforward to design an explore-then-commit style learning algorithm that achieves logarithmic regret.
>
> So, we agree that the explore-then-commit paradigm and using UCB/LCB estimates are standard. However, the commit phase of the learning algorithm relies on the MDP policy, and our main contribution is the novel MDP policy and proving that it has constant regret.
>
> ## Response to question 3
> [Similar response to question 2 from reviewer S3ZC]
> Some experimental results are now included in the rebuttal revision in Appendix E.
>
> We acknowledge that our learning algorithm is not very practical because of the constant factors, although using the empirical means instead of the true means seems to work on simple test cases. However, the main contribution of our work is designing a novel MDP policy, ControlBudget, and showing that it has constant regret. This also lets any future learning algorithms to be compared directly to the LP instead of the optimal policy because, as our theorems in section 3 show, the gap between the optimal policy and LP is a constant.

---

### Official Review · Reviewer_5cwS · 2022-07-20

**Rating:** 6
**Confidence:** 2
**Soundness:** 3 good
**Presentation:** 2 fair
**Contribution:** 3 good

**Summary:**

The paper introduced a generalization of BwK model which admits increasing in budget. The authors then proposed an offline MDP-based algorithm named "CB", follows by a learning algorithm that combines the offline version with Li et.al's algorithms. The regret analysis of proposed algorithms is provided and it is tight. Interestingly, the offline algorithm achieves constant regret against OPT_LP, implying the gaps between the expected total reward of "CB", OPT and OPT_LP are not larger than a constant. Besides, the regret of the learning algorithm matches the result in [14] when applying to the classic BwK model.

**Questions:**

I have 3 following questions:

Q1: While the problem is well motivated as the resources can be renewed, the proposed model is rather unnatural. Particularly, the assumption that both consumption and replenishment depend on arm pulled and are represented by only one variable does not sound reasonable to me. Moreover, I understand that the notion of the null arm is technical and only serves for the analysis. Is there any practical motivation to introduce this arm into the model?

Q2: I found the proof of lemma 3.1 and its generalization (line 209) are not clear. Which properties of LPs and case analysis are used? Perhaps it is based on the assumption that |X*| =|J*| as in [14]? I do think the authors should be more specific to avoid confusion here.

C3: What is the advantage of the offline MDP-based algorithm (Algorithm 2) over solving the LP (1), as the algorithm also requires solving linear programming within its procedure?



**Ethics Review Area:**

["I don’t know"]

**Strengths And Weaknesses:**

Pros:
The introduced model is new and can serve as a generalisation of BwK model. Accordingly, the two proposed algorithm is new and the  regret is tight.

The offline method and its analysis is new and insightful.

Cons:
The paper sometimes lacks clarity and I found the its organisation is not easy to follow.

---

> ### Author Response · Authors · 2022-08-02
> **Response**
>
> Thank you for your feedback. Here are the responses to your questions.
>
> ## Response to question 1
> [Related to response to question 1 from reviewer S3ZC]
> Our presentation is in terms of drifts instead of replenishment and consumption separately only for notational convenience. Our model incorporates models in which resource replenishment and consumption are separate stochastic processes, simply by defining the drift $d_t$ in our model to be the difference of the replenishment and consumption vectors produced by the arm chosen in round $t$. Our assumption that both replenishment and consumption may be stochastic, and may depend on the arm chosen, is not a limiting assumption since it includes the special cases when the amount of resource replenishment or consumption (or both) does not depend on the arm chosen, or when the replenishment or consumption process (or both) is deterministic. As far as we are aware, the only one of these special cases that simplifies the problem significantly is when both replenishment and consumption are deterministic and the expected reward of each arm is known. In that case, since the state evolution is deterministic, the problem reduces to a path planning problem that can be solved using the Bellman-Ford algorithm.
>
> ## Response to question 2
> Since we assume uniqueness of the LP solution, similar to assumption 2 of [14], our claim follows from lemma 1 of [14]. Since we include the sum to one constraint in the LP, we have $|X^*| + |J'| = m+1$, which implies that $|X^*| \leq m+1$ (this is equal to 2 in the one resource case).
>
> In the one resource case, consider all possible solutions that are supported on at most two arms (e.g., null arm plus negative drift arm, positive drift arm plus negative drift arm, etc.). One way of checking which ones are valid is to instead consider which LP constraints are binding. There are $n+2$ constraints - $n$ nonnegativity constraints, 1 sum-to-one constraint, and 1 resource constraint. Since the LP solution is supported on one or two arms, $n$ or $n+1$ constraints are binding. For example, if $n-1$ nonnegativity constraints are tight and the sum to one constraint is tight, then the solution must be supported a single arm and this arm must have positive drift - this follows from the slack in the resource constraint and our assumption that $T \geq \frac{B}{\delta_{\text{drift}}}$. Continuing in this way yields that the LP solution is one of the three types listed in the paper. Note that we make the assumption $T \geq \frac{B}{\delta_{\text{drift}}}$ only in the one resource case in this section to simplify the cases and focus on providing intuition behind our algorithm.
>
> ## Response to question 3
> The advantage of the offline MDP-based algorithm (Algorithm 2) is that its regret is asymptotically better than the regret obtained by solving LP (1) and pulling arms drawn at random from the LP-optimal distribution $p^*$. As Flajolet and Jaillet [9] show, the policy of sampling arms at random from $p^*$ leads to $\Omega(\sqrt{T})$ regret, whereas our policy leads to $O(1)$ regret.
>
> The improvement in regret can be attributed to the following feature of Algorithm 2. Instead of merely sampling arms from the optimal LP solution, our policy samples an arm from a probability distribution obtained by solving a **perturbed** version of the LP. Specifically, the right-hand side of each constraint is perturbed to account for over/under consumption of resources, ensuring that the budget of each resource stays close to a decreasing sequence of thresholds. The sequence is chosen such that the expected leftover budget is a constant and proving this a key step in the regret analysis. Our learning algorithm uses this as a key component and achieves $O(\log T)$ regret that is exponentially better than the $O(\sqrt{T})$ regret obtained by solving the LP (1).

---

### Meta-Review · Area_Chair_zQzM · 2022-08-26

**Recommendation:** Accept
**Confidence:** Certain

**Metareview:**

The paper investigates a generalisation of the BwK problem where the budget is increasing. The authors first propose an offline MDP-based algorithm called "CB”, and an online learning algorithm that combines the offline version with an algorithm from Li et.al. The authors also provide a regret analysis for the proposed online algorithm and they show that it is tight. Interestingly, the offline algorithm achieves constant regret against OPT_LP, implying the gaps between the expected total reward of "CB", OPT and OPT_LP are not larger than a constant. This is probably due to the fact that the pulling costs are bounded (which matches with the known results from the knapsack literature). The authors also mention that the regret of the learning algorithm matches the result in [14] when applying to the classic BwK model.

After the rebuttal phase, all the reviewers are happy with the responses, and suggest accepting this paper.

**Award:**

No

---

### Decision · Program_Chairs · 2022-09-14

Accept